# A social system to disperse the irrigation start date based on the spatial public goods game

**Yoshiaki Nakagawa** [ID] * , **Masayuki Yokozawa** [ID]

Faculty of Human Sciences, Waseda University, Mikajima, Tokorozawa, Japan

* nrj59355@nifty.com

## Abstract

In paddy rice cultivation, the amount of water used during the beginning of the irrigation season is the highest. However, there is a possibility of a water shortage at this season as climate change decreases snowfall. In this study, we propose new schemes based on the public goods game to reduce peak water volume during this season by dispersing the irrigation start dates. In our agent-based model, agents determine the irrigation start date based on the evolutionary game theory. This model considers the economic variables of individual farmers (e.g., gross cultivation profit and cultivation cost), the cost and subsidy for cooperation for the dispersion of the irrigation start dates, and the information-sharing network between farmers. Individual farmers update the cooperation/defection strategy at each time step based on their payoffs. Using this agent-based model simulation, we investigate a scheme that maximizes the dispersion of irrigation start dates among multiple scheme candidates. The results of the simulation show that, under the schemes in which one farmer belongs to a group and the groups do not overlap, the number of cooperating farmers did not increase, and the dispersion of irrigation start dates barely increased. By adopting a scheme in which one farmer belongs to multiple groups and the groups overlap, the number of cooperating farmers increased, while the dispersion of irrigation start dates maximized. Furthermore, the proposed schemes require the government to obtain information about the number of cooperators in each group to determine the subsidy amount. Therefore, we also proposed the method which allows estimating the number of cooperators in each group through the dispersion of irrigation start dates. This significantly reduces the cost of running the schemes and provides subsidization and policy evaluations unaffected by false declarations of farmers.

## 1. Introduction

Rice is one of the most important staple food crops and is widely cultivated especially in Asia region to feed approximately 3 billion people [1]. Seventy-five percent of rice paddies are irrigated, which requires a lot of fresh water [2]. The rice cultivation season in Japan is generally from May to September, with irrigation from May to August. In May, field is flooded by irrigation water and rice is transplanted. Specifically, the operation called puddling in rice paddies

**Funding:** Initials of the authors who received each award: MY Grant numbers awarded to each author: JPMEERF20S11805 The full name of each funder:

the Environment Research and Technology Development Fund (the Environmental Restoration and Conservation Agency of Japan) URL of each funder website: https://www.erca.go.jp/erca/english/index.html Did the sponsors or funders play any role in the study design, data collection and analysis, decision to publish, or preparation of the manuscript?: The funders had no role in study design, data collection and analysis, decision to publish, or preparation of the manuscript.

**Competing interests:** The authors have declared that no competing interests exist.

at the beginning of the irrigation season is most important for the rice cultivation. Puddling softens the soil, levels the surface of the field, and reduces permeability, requiring large amounts of water during the puddling stage [3, 4]. In recent years, however, there has been concern about water shortages in rice paddies. The reasons are population growth, water use conflicts with the non-agricultural sector, and changes in water resources due to climate change [5] and so on. Future climate change may reduce the amount of snowfall, thereby decreasing meltwater and reducing river flow, which in turn can cause drought during the puddling stage [6, 7]. Therefore, to adapt to climate change, it is necessary to have measures to prevent water shortages during this season by dispersing irrigation start dates among farmers and minimizing the peak water consumption (maximum water intake). Traditionally, paddy rice farmers in several Asian regions have formed groups such as irrigation associations with neighboring farmers, and these groups are responsible for managing agricultural water [8–14]. Therefore, as a concrete measure to minimize peak water consumption, the government could provide subsidies to these groups based on the number of farmers cooperating to disperse their irrigation start dates. This study examines subsidy policies and effective group building to reduce peak water consumption during this period by dispersing irrigation start dates with as few subsidies as possible.

This measure can be regarded as a kind of "public goods game" in which the government pays subsidies to the entire group according to the cooperation costs incurred by the cooperative farmers within the group. In general, the public goods game is played according to the following rules. Cooperators contribute cost to the public good; defectors do not contribute. The total contribution is multiplied by a multiplication factor $r$, and the result (i.e., payoff derived from the public good) is evenly divided among all members. In the measure mentioned above, subsidies can be regarded as the payoff derived from the public good, and because the amount of subsidy is determined according to the number of cooperators (or the cost of cooperation). Furthermore, the public goods game introduced static relationships among agents such as grids and networks is called the spatial public goods game (e.g., [15, 16]). In this study, we assumed that farmers refer to strategies and payoffs of other farmers in their geographical proximity. Therefore, we introduce an information reference network of farmers into the public goods game. This kind of public goods game is called the spatial public goods game. There is extensive research on spatial public goods games in physics, mathematics, information science, and economics [17, 18]. Accordingly, the extensive body of research on spatial public goods games to date can be used as a basis to design social schemes in this study.

In previous studies on the spatial public goods games, it is known that the degree of promotion of cooperation varied depending on a pre-setup for internal groups in the player population [17, 19]. In most studies on spatial public goods games, one agent and adjacent agents (i.e., von Neumann neighborhood) on the grid are regarded as one group [17, 20–24]. In this group style, the groups overlapped (Fig 1B). However, agricultural water is often managed and adjusted by non-overlapped (or completely divided) groups as shown in Fig 1A (e.g., water user associations such as the land improvement districts in Japan, Subak in Bali, Indonesia, Zanjera in Philippine, and Muang Fai in Thailand; [10, 11, 25–27]). Therefore, we examined and compared the results of the spatial public goods game between overlapped and completely divided groups.

Furthermore, in the basic pre-setup for internal groups in the spatial public goods game, agents belong to multiple neighborhood groups [17, 20, 23, 24]. For instance, a focal agent belongs to a group of five given four adjacent agents (In Fig 1B, the focal agent is represented by a black closed circle, and the five groups are represented by five diamond-shaped frames). Payoffs are calculated based on the number of cooperators for each group to which the focal agent belongs, and the focal agent obtains the total of these payoffs. However, in a simpler

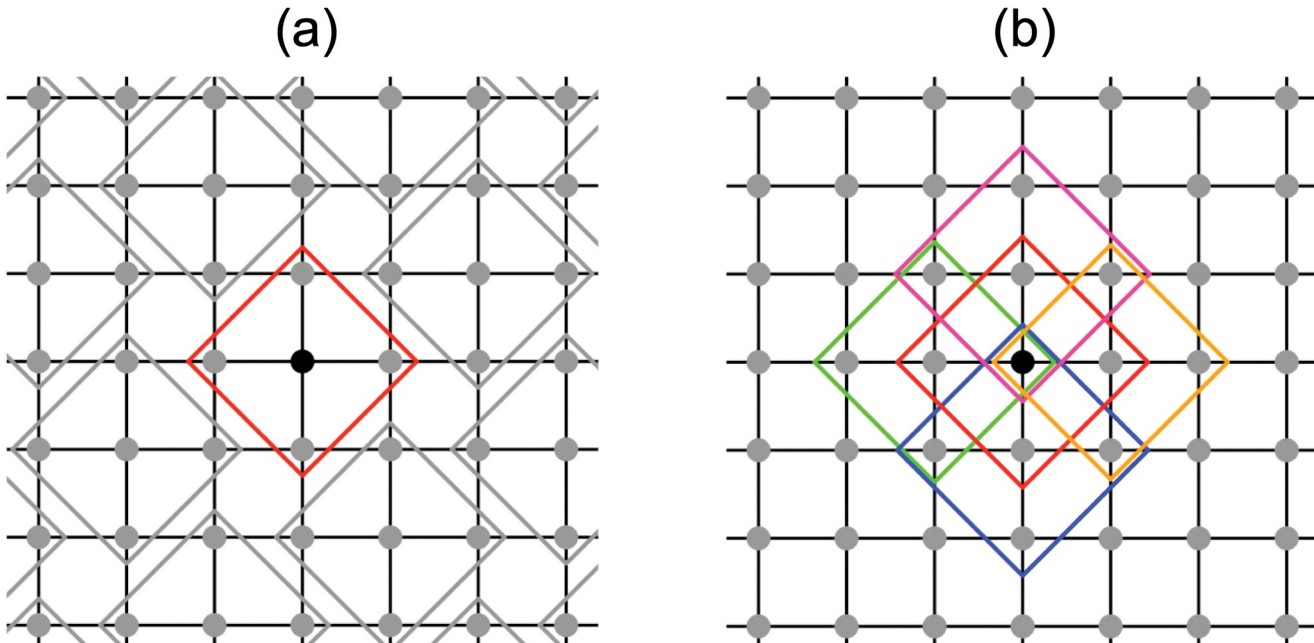

**Fig 1. A schematic diagram of a group in a spatial public goods game.** Panels (a) and (b) show a divided (non-overlapped) group and an overlapped group, respectively. The grid points represent the farmers, and the black grid points represent the target farmer. The diamond-shaped frame represents a group. Panel (a) corresponds to Group Condition 1, and the red frame represents the group to which the target farmer and neighboring farmers belong. The gray frames represent the groups of other farmers. Panel (b) corresponds to Group Conditions 2–4, and for ease of viewing, only five groups to which the target farmer belongs are drawn as different-colored frames.

version of this, the focal agent gains the payoff only depending on the number of cooperators in one group to which the focal agent and its neighboring agents belong [21]. Therefore, in this study, we also examined and compared the results of the spatial public goods game when the focal agent obtains payoffs calculated by one group and by multiple groups.

Furthermore, as another pre-setup, we define two types of groups: a group for farmers to discuss and adjust the irrigation start dates and a group for the government to calculate subsidies based on the dispersion of these dates. Many studies on spatial public goods games assumed that the two groups are the same [17, 21–24], but in this study, we examined not only the outcomes of having the same group sizes, but also those of having different group sizes. This is because, in practice, the size of groups assumed by the government is not always the same as that assumed by farmers.

Uncertainty of variables and parameters must be considered when applying game theory to creating real-world institutions [28]. Ostrom [28] pointed out that the central government's inability to fully obtain information about the players disrupts the control of common-pool resources. As such, it is difficult to achieve cooperation predicted by game theory, unless the central government accurately grasps the values of several parameters and variables such as the strategy adopted by the player (cooperation/defection), payoff, and cost. In addition, considerable administrative costs are involved in grasping the values of such parameters and variables completely and precisely. Similarly, in the measures proposed in this study, the government must obtain information on the number of cooperating and defector farmers in each group to calculate the subsidy amount. If all farmers self-report their strategies (cooperation or defection) to the government and the self-report is credible, the government can accurately capture information on these numbers. However, self-reporting can often be misleading, as farmers are more likely to report themselves as cooperators than defectors to the government when

they choose to defect. To address this problem, we propose a method for governments to obtain farmer player information at low administrative costs. In the practical implementation of this method, the government estimates the number of cooperators within a group based on the dispersion of irrigation start dates. In recent years, the development of satellite technology has enabled the government in monitoring irrigation start dates at the level of individual paddy fields [4, 29–32]. The dispersion of irrigation start dates within groups can be obtained from this satellite data. Furthermore, we examined the effect of uncertainty in the estimated value of the number of cooperators within a group when using this method on outcomes of the number of cooperators and the dispersion of the irrigation start date. This is because the group condition that most promotes cooperation under complete information conditions may be not always the same as that under incomplete information conditions (i.e., the condition in which the estimates of the number of cooperators/defectors have uncertainties).

In the basic model of this study, we assumed that farmers determine the irrigation start date based on payoff, but in real-life scenarios, farmers may prefer to determine a particular day as the irrigation start date due to factors other than payoff (e.g., cultural and social factors such as long-standing traditions and religious beliefs). It has been seen that irrigation start days tend to be concentrated on Saturdays and Sundays because of the increasing number of farmers with side jobs in Japan and the irrigation start days tend to overlap due to the designation of planting times by agricultural cooperatives and extension workers [31]. Therefore, we also examined the impact of the proposed schemes on the diffusion of cooperators and dispersion of the irrigation start dates, among farmers with this exogenous preference.

In this study, we demonstrate the effectiveness of recent theoretical developments in spatial public goods games in combination with the distribution of irrigation start dates obtained from satellite data, to design effective adaptation measures.

## 2. Methods (model and parameters)

### 2.1. Outline of simulation

The basic simulations of spatial public goods games in this study were performed in a periodic square lattice [24]. Each site on the lattice was occupied by a farmer. It assumes that the farmers are uniformly distributed on a two-dimensional plane, and each farmer is linked to (or shares information with) the four geographically closest farmers (i.e., the von Neumann neighborhood). There were 900 farmers, who were placed on grid points of a $30 \times 30$ two-dimensional square lattice. Initially, the two strategies (cooperation or defection) were set to be in the same proportion and were randomly and evenly distributed throughout the grid.

First, farmers are divided into multiple groups. Every year, they decide whether to cooperate in dispersing the irrigation start dates based on their strategies (Fig 2A). Cooperating farmers consult with other farmers in the same group to promote the dispersion of irrigation start dates and determine their irrigation start dates. In contrast, defecting farmers choose the optimal irrigation start date to maximize yield and sales (Fig 2B). In scenarios 1 and 2, the farmer does not have an extrinsic preference for irrigation start dates, which is explained in the introduction, but in scenario 3 the farmer does. Every year, the player farmer grew rice, and earned income (sales minus cultivation cost) (Fig 2C). Cooperating farmers often fail to select the optimal irrigation start date that maximizes yield and sales, resulting in reduced yields and sales, thus incurring cooperation costs (Fig 2C).

Subsequently, the government obtains information on the number of cooperating and defecting farmers in each group (Fig 2F). In scenario 1, farmers truthfully report their strategies to the government, therefore, the government has accurate information regarding the number of cooperating and defecting farmers in each group (Fig 2D: complete information

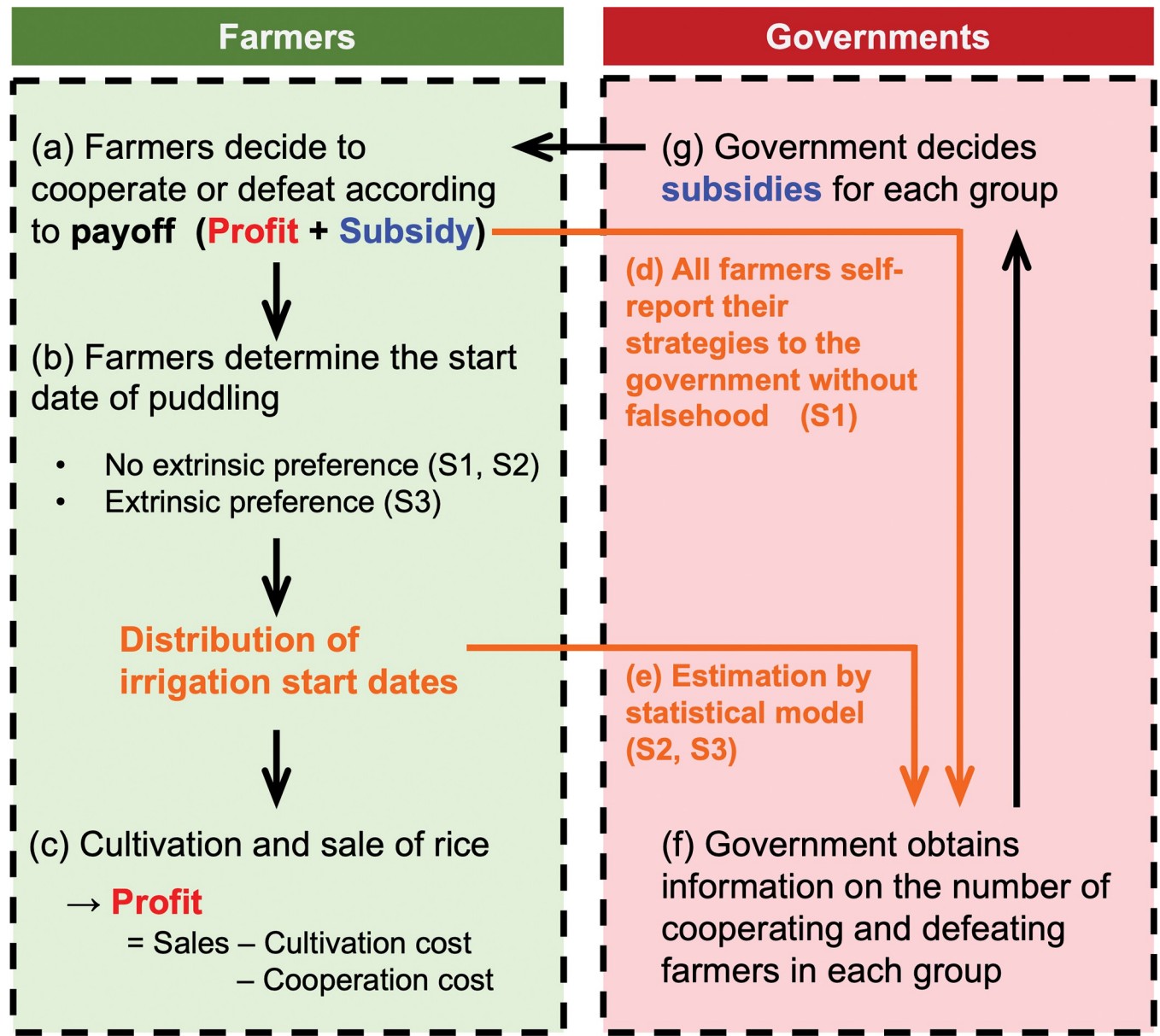

**Fig 2. Schematic diagram of the proposed model for dispersing the irrigation start dates.** S1, 2, and 3 represent the condition of complete information without exogenous preference (scenario 1), the condition of incomplete information without exogenous preference (scenario 2), and the condition of incomplete information with exogenous preference (scenario 3), respectively.

condition). Although this scenario is an unrealistic setting, it was run to compare the results of this scenario with those of more realistic scenarios (scenarios 2 and 3). In scenarios 2 and 3, the government estimated the number of cooperating and defecting farmers in each group from the distribution of irrigation start dates data using statistical models (Fig 2E: incomplete information condition). The government finally decides on subsidies for each group of farmers based on the information about the number of cooperating and defecting farmers in each group, and subsidizes each group of farmers (Fig 2G). The subsidy amount increases as the number of cooperative farmers within the group increases. The subsidy is equally divided among the farmers in the group. There are four subsidy calculation schemes (or group conditions), which are detailed in Section 2.2.

At the end of each year, the player farmer referenced the payoff (income plus subsidy) of the adjacent player farmer in the square lattice, and stochastically decided whether to mimic the strategy of an adjacent player farmer based on their payoffs. Under each condition (i.e., the group conditions, external preference conditions, parameter conditions, etc., which are explained in the following sections), a 200-year simulation was run, and 100 trials were repeated.

The "Keep It Simple, Stupid" (KISS) principle put forth by Robert Axelrod is a useful guideline to model an agent-based simulation [33]. The agent-based model in this study followed this guideline and excluded some elements of the target phenomenon, in the interest of simplicity.

## 2.2. Group conditions

In this study, each group consisted of five farmers, and the simulation was performed for four group conditions. There was no overlap in Group Condition 1 (Fig 1A), but there was an overlap in Group Conditions 2–4 (Fig 1B).

In addition, the group for farmers who discussed and adjusted the irrigation start dates and the group for the government to calculate subsidies were defined separately. Under Group Condition 1, it was defined that these two types of groups were the same group and the number of groups of each type was one for each farmer (red frame in Fig 1A). Under Group Condition 2, it was defined that both types of groups were the same group, the number of groups of each type was one for each farmer, and each farmer was located at the center in the group (red frame in Fig 1B). Under Group Condition 3, it was defined that both types of groups were the different group, the number of groups to discuss and adjust the irrigation start date was five for each farmer (all frames in Fig 1B), and the number of groups to calculate the subsidy was one for each farmer (red frame in Fig 1B). Therefore, the size of former type of group was larger than that of latter type of group. In this condition, farmers are more active in dispersing the irrigation start dates and discuss and adjust irrigation start dates with more farmers, even if their discussion and adjustment are not subject to subsidy assessment. Under Group Condition 4, it was defined that both types of groups were the same group and the number of groups of each type was five for each farmer (all frames in Fig 1B).

## 2.3. Payoff calculation modules

**(i) Payoff calculation for Group Condition 1.** The payoff when farmer $i$ is a defector is described by the following equation:

$$M_{i,D} = R_{opt} - C_{opt} + \frac{C_{co}rn_c}{k_i + 1} \tag{1}$$

where $R_{opt}$ and $C_{opt}$ represent the rice sales [yen/ha/year] and the cost of rice cultivation [yen/ha/year], respectively, under the condition of the optimal irrigation start date; $C_{co}$ is the cooperation cost (i.e., the cost due to the loss of yield and sales due to late or early irrigation start dates and planting) [yen/ha/year]; $n_c$ is the number of cooperators in the group to which farmer $i$ belongs; $k_i$ is the degree of farmer $i$ (i.e., the total number of links connected to farmer $i$, that is, 4). The optimal irrigation start date is the most suitable irrigation start date for rice cultivation. When farmers choose the optimal irrigation start date, they can maximize their income because of higher yield and sales or lower cultivation costs. $C_{co}rn_c$ is the subsidy for the group [yen/ha/year]; therefore, the third term of the equation is the subsidy amount received by farmer $i$. In addition, in previous studies on public goods games, $r$ is a parameter called the multiplication factor (or enhanced factor). In this study, $r$ is the subsidy coefficient set by the

government. The greater the subsidy coefficient, the greater the subsidy amount. Moreover, based on "Management income and expenditure of agricultural management entity in 2019" by the Japanese Ministry of Agriculture, Forestry and Fisheries [34], the rice sale and cost of rice cultivation under the optimal irrigation start date condition were set as 13,94,476 [yen/ha/year] and 12,63,050 [yen/ha/year], respectively. We examined the cooperation cost under the following conditions: $C_{co} = m \times 10000$[yen/ha/year], $(m = 1, \ldots, 6)$. The payoff when farmer $i$ is a cooperator is described by the following equation:

$$M_{i,C} = \begin{cases} M_{i,D} - C_{co}(\text{nonoptimal irrigation start date condition}) \\ M_{i,D}(\text{optimal irrigation start date condition}) \end{cases} \quad (2)$$

The cooperators with the optimal irrigation start date conditions pays the cost due to the loss of yield and sales due to late or early irrigation start dates and planting as the cooperation cost. The dispersion of the irrigation start date is not maximized if all cooperators select the nonoptimal irrigation start date. For example, if all farmers are cooperators and all cooperators select the nonoptimal irrigation start date, no farmer select the optimal irrigation start date. Therefore, in this study, when there are few farmers who select the optimal irrigation start date, cooperators select this date. The cooperators with the nonoptimal irrigation start date conditions do not pay the cooperation cost. In this study, they did not pay the cooperation cost, but they are counted as cooperators in the subsidy calculation, because they contribute to increase the dispersion of the irrigation start date.

Eqs (1) and (2) are modified for farmers by adding rice sales and cultivation costs to the formulas used in a previous study on spatial public goods games (e.g., [35]).

**(ii) Payoff calculations for Group Conditions 2 and 3.**   The payoff when farmer $i$ is a defector is described by the following equation [21]:

$$M_{i,D} = R_{opt} - C_{opt} + \frac{C_{co}rn_{c,i}}{k_i + 1} \quad (3)$$

where $n_{c,i}$ is the number of cooperators in the group in which farmer $i$ is at the center (i.e., the number of cooperators in the group consisting of farmer $i$ and four farmers in the von Neumann neighborhood of farmer $i$). The payoff when farmer $i$ is a cooperator is described by the following equation:

$$M_{i,C} = \begin{cases} M_{i,D} - C_{co}(\text{nonoptimal irrigation start date condition}) \\ M_{i,D}(\text{optimal irrigation start date condition}) \end{cases} \quad (4)$$

**(iii) Payoff calculation for Group Condition 4.**   When farmer $i$ is in the von Neumann neighborhood of farmer $j$, the subsidy of farmer $i$ with strategy $s_i$ in the group centered on farmer $j$ is described by the following equation [17, 20, 23]:

$$S_{i,j} = \frac{r}{k_j + 1} \sum_{m=0}^{k_j} \frac{C_{co}}{k_m + 1} s_m - \frac{C_{co}}{k_i + 1} s_i \quad (5)$$

where $k_i$ represents the degree of farmer $i$; when the strategy of farmer $i$ is cooperation, $s_i = 1$; when the strategy of farmer $i$ is defection, $s_i = 0$; $m = 0$ stands for farmer $j$. Therefore, the payoff

when farmer $i$ is a defector is described by the following equation:

$$M_{i,D} = R_{opt} - C_{opt} + \sum_{j \in \Omega_i} S_{i,j} = R_{opt} - C_{opt} + \sum_{j \in \Omega_i} \left( \frac{r}{k_j + 1} \sum_{m=0}^{k_j} \frac{C_{co}}{k_m + 1} s_m \right) \qquad (6)$$

where $\Omega_i$ is the set of farmers that are in the von Neumann neighborhoods of farmer $i$. The payoff when farmer $i$ is a cooperator is described by the following equation:

$$M_{i,C} = \begin{cases} M_{i,D} - C_{co}(\text{nonoptimal irrigation start date condition}) \\ M_{i,D}(\text{optimal irrigation start date condition}) \end{cases} \qquad (7)$$

## 2.4. Strategy update rules

Farmers update their strategy after their payoffs have been determined. First, the target farmer $i$ randomly selects a reference farmer from all farmers linked to him/herself on a square lattice [17]. In addition, farmer $i$ follows the strategy of farmer $j$ (i.e., the reference farmer) with the following probabilities [23, 24]:

$$p(i \rightarrow j) = \frac{1}{1 + exp[(M_i - M_j)/\kappa]} \qquad (8)$$

where $\kappa$ is the magnitude of noise. Based on previous studies [23, 36], we set $\kappa = 0.1$ in this study.

## 2.5. Irrigation start date determination module

The player farmer determines the irrigation start date using either of the following two methods: As defecting farmers do not cooperate in the dispersion of the irrigation start date, they choose the optimal irrigation start date. Because the cooperating farmers cooperate in dispersing the irrigation start dates, they decide their irrigation start dates by selecting from dates least selected by other farmers (i.e., the defecting and cooperating farmers of the same discussion and adjustment group, who have already selected their irrigation start dates). In this study, the number of classes of irrigation start dates was set to 15, and the optimal irrigation start date was set to the 8th class, which was located in the center of the entire period. As the number of classes increased, the time scale resolution increased.

**(i) Irrigation start date determination module without exogenous preferences.**

1. All defecting farmers choose the optimal irrigation starting date as their irrigation starting date.

2. We randomly select one cooperating farmer.

3. The selected cooperative farmer then selects a day class least frequently selected by farmers belonging to the same group as themselves as their irrigation start date. If there are multiple day classes least frequently selected by farmers belonging to the same group as the selected cooperative farmer, the selected cooperative farmer randomly selects a day class from among them as their irrigation start date. If these farmers select the optimal irrigation start date the least frequently, then that date is set to the irrigation start date of the selected coopering farmer.

4. Steps 2 and 3 were repeated until the irrigation start date for all cooperating farmers was determined.

**(ii) Irrigation start date determination module with exogenous preferences.** In addition, in the irrigation start date determination module (ii), we introduce an exogenous preference for irrigation start dates. Here, the exogenous preference is due to factors different from the payoff considered in the agent-based model of this study, for example, cultural and social factors, such as long-standing traditions and religious beliefs. Specifically, cooperating farmers preferentially select the day closest to the optimal irrigation start date from the days selected as the irrigation start date due to exogenous preferences.

1. All defecting farmers choose the optimal irrigation starting date as their irrigation starting date.

2. We randomly select one cooperating farmer.

3. The selected cooperative farmer then selects a day class least frequently selected by farmers belonging to the same group as themselves as their irrigation start date. If there are multiple day classes least frequently selected by farmers belonging to the same group as the selected cooperative farmer, the selected cooperative farmer selects a day class closest to the optimal irrigation start date among them as their irrigation start date. If these farmers select the optimal irrigation start date the least frequently, then that date is set to the irrigation start date of the selected coopering farmer.

4. Steps 2 and 3 were repeated until the irrigation start date for all cooperating farmers was determined.

## 2.6. Module for estimating the number of cooperators

**(i) Complete information condition.** Under this condition, the government can obtain accurate information on the number of cooperating farmers (and defecting farmers) in each group and calculate subsidies based on that information. In this case, in which the government has a very high ability to monitor all farmers, or all farmers report their strategy (cooperation or defection) to the government honestly. As an implementation, the true value of the number of cooperating farmers in each group was used in the subsidy calculation in the payoff calculation modules (Section 2.3).

**(ii) Incomplete information condition.** Under this condition, the government estimates the number of cooperators in each group from the unevenness of the irrigation start dates. The degree of unevenness for all farmers belonging to group $i$ for the government to calculate subsidies is described by the following equation:

$$U_i = \sum_{k=1}^{N_{day}} p_k{}^2 \tag{9}$$

where $N_{day}$ is the number of day classes that can be selected as the irrigation start date, and $p_k$ is the number of farmers who select the $k$th day class as the irrigation start date. In this study, $N_{day}$ was set to 15. The relationship between $U_i$ and the number of defectors belonging to group $i$, $n_{d,i}$, is approximated by the following equation:

$$n_{d,i} = round(b + a \cdot ln(U_i)) \tag{10}$$

where $a$ and $b$ are tuning parameters, and $round(x)$ is a function that converts $x$ to the nearest neighbor integer. The values of a and b were estimated by performing a regression analysis of the simulation data using the following equation: $n_{d,i} = b + a \cdot ln(U_i)$. The results showed that the estimates for $a$ and $b$ were 2.431 and 4.988, respectively. The coefficient of determination ($R^2$) is 0.998.

## 2.7. Changes in simulation results for inter-farmer information reference networks of different average degrees

In the above basic simulation, the information reference network of farmers is assumed to be a square lattice, i.e., the degree of each farmer is four. Based on this network structure assumption, farmers refer to the payoffs of other farmers that are randomly selected from among four neighboring farmers linked to himself/herself. Empirical studies have reported that the average degree of reference networks for farmers' agricultural techniques and information ranges from approximately 4 to 19 [37–39]. Therefore, in this study, we also investigated the simulation results under the conditions of an information reference network with an average degree other than four. We created networks of average degrees 8, 12, and 20 and performed simulations using these networks without changing the other simulation conditions. In doing so, we assumed that farmers were uniformly distributed and shared information with those who were geographically closer, as in the basic simulation. Fig 3 shows the link relationships between farmers under each average-degree condition.

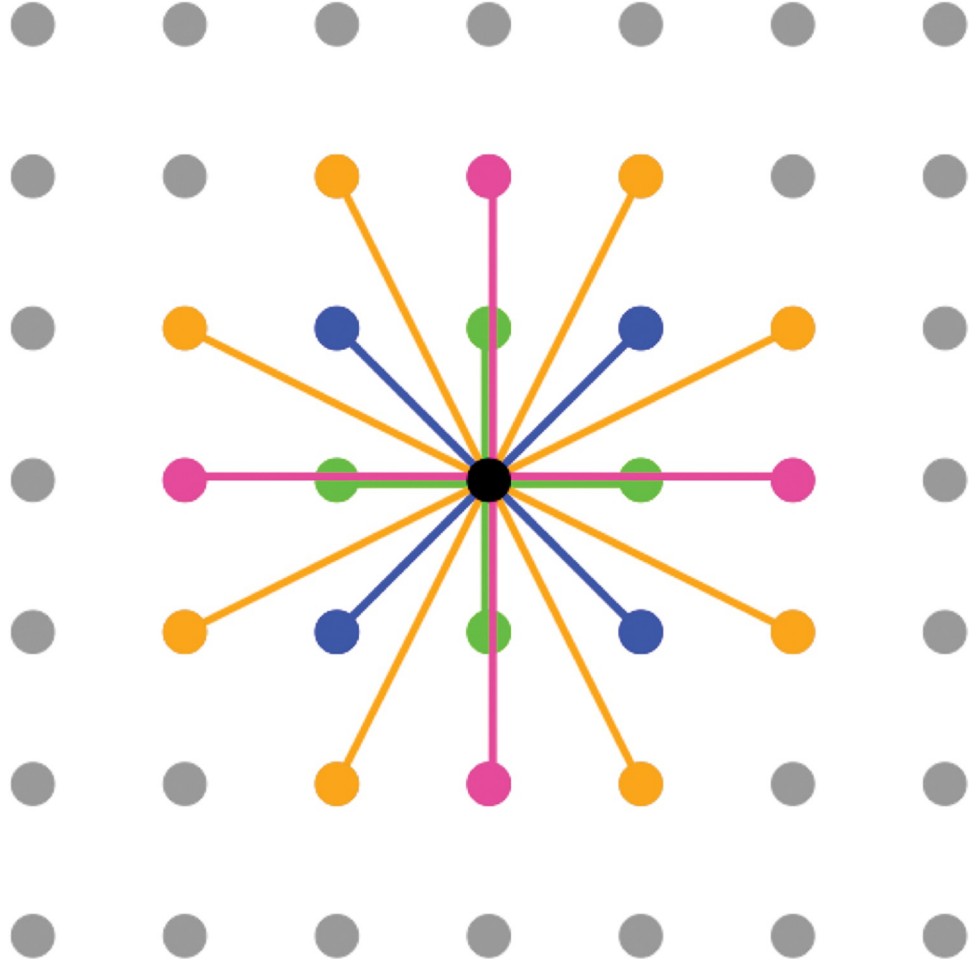

**Fig 3. Links between focal farms and other farms in various degree conditions.** The closed circles indicate farmers and are uniformly distributed on a two-dimensional plane. The black circles indicate focal farmers. A straight line connecting the circles indicates a link between the farmers. Under the condition in which the average degree is four, the green straight lines represent the links between the farmers. Under the condition in which the average degree is eight, the green and blue straight lines represent the links between the farmers. Under the condition in which the average degree is 12, the green, blue, and pink straight lines represent the links between the farmers. Under the condition where the average degree is 20, the green, blue, pink, and yellow straight lines represent the links between the farmers.

## 3. Results

We performed simulations under the condition that the cooperation cost, $C_{co}$, is 10000–100000 [yen/ha/year]. However, there was little difference in the results due to cooperation costs (see Section 3.4 for details). Therefore, in this study, the result for $C_{co} = 10000$ is shown below.

### 3.1. Condition of complete information without exogenous preference

Under this condition with the overlapped group condition, as $r$ increased, the proportion of cooperators increased, and at $r > 5$, all players became cooperators (panels in the first row of Fig 4). Meanwhile, in the divided group condition, when $r \leq 5$, the proportion of cooperators increased as $r$ increased, but when $r > 5$, it did not increase as $r$ increased (panels in the first row of Fig 4). The percentage of cooperators at the time was less than 20%. Therefore, the overlapped group condition promoted cooperation more than the divided group condition did.

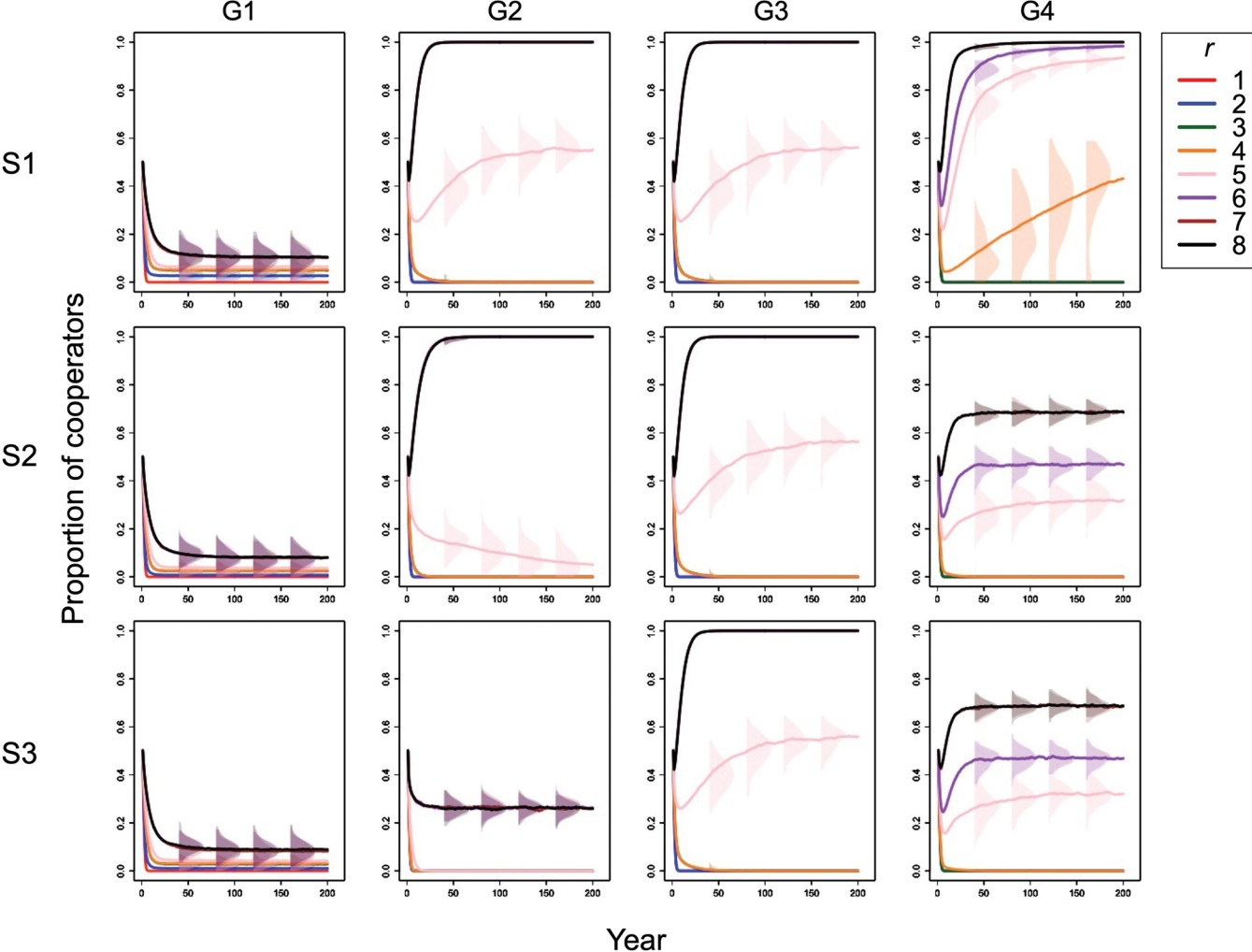

**Fig 4. Temporal change in the proportion of cooperators.** Columns (G1)–(G4) of the panel show the results for Group Conditions 1–4. Rows (S1)–(S3) of the panel display the condition of complete information without exogenous preference (scenario 1), the condition of incomplete information without exogenous preference (scenario 2), and the condition of incomplete information with exogenous preference (scenario 3), respectively. The color of line indicates condition $r$. Colored areas represent the distributions of the cooperator proportions for multiple trials.

The proportion of cooperators in Group Condition 4 was much greater than those in Group Conditions 2 and 3 at $r = 4$ and 5 (panels in the first row of Fig 4). Therefore, Group Condition 4 was the most conducive to cooperation among all overlapped group conditions.

Furthermore, as the proportion of cooperators increased, the dispersion of the irrigation start date also increased (Fig 5). Therefore, the overlapped group condition promoted the dispersion of the irrigation start date more than the divided group condition did. In addition, Group Condition 4 had the greatest dispersion of the irrigation start date among all the overlapped group conditions.

### 3.2. Condition of incomplete information without exogenous preference

Even in this condition, the overlapped group condition promoted cooperation more than the divided group condition did (panels in the second row of Fig 4).

At $r > 5$, the proportion of cooperators in Group Condition 4 was smaller than those in Group Conditions 2 and 3 (panels in the second row of Fig 4). At $r = 5$, the proportion of cooperators in Group Condition 3 was larger than that in Group Condition 2. Therefore, Group Condition 3 was the most conducive to cooperation among all overlapped group conditions.

Even under this condition, as the proportion of cooperators increased, the dispersion of the irrigation start date also increased (Fig 6). In addition, Group Condition 3 had the greatest dispersion of the irrigation start dates among all the overlapped group conditions.

The reading of the figure is the same as that in Fig 5.

### 3.3. Condition of incomplete information with exogenous preference

Even in this condition, the overlapped group condition promoted cooperation more than the divided group condition did (panels in the third row of Fig 4).

In Group Condition 2, all players became defectors at $r = 5$, and at $r > 5$, the proportion of cooperators did not increase from approximately 30%, even if $r$ increased (panels in the third row of Fig 4). Therefore, at $r > 5$, the proportion of cooperators in Group Condition 2 was smaller than that in Group Conditions 3 and 4. Therefore, Group Condition 2 was the least conducive to cooperation among all the overlapped group conditions.

In addition, at $r > 5$, the proportion of cooperators in Group Condition 3 was larger than that in Group Condition 4 (panels in the third row of Fig 4). Therefore, Group Condition 3 was the most conducive to cooperation among all overlapped group conditions.

Under this condition, due to the exogenous preference, the irrigation start dates tended to concentrate to the optimal irrigation start date even when $r$ was relatively large (Fig 7). Even under this condition, as the proportion of cooperators increased, the dispersion of the irrigation start dates also increased. Among the overlapped group conditions, the dispersion of the irrigation start date increased in Group Conditions 3, 4, and 2, in that order.

The reading of the figure is the same as that in Fig 5.

### 3.4. Robustness of simulation results against changes in the cost amount

The results of the proportion of cooperators and the dispersion of the irrigation start dates within the population remained almost unchanged regardless of the change in the cooperation cost. Specifically, the relationships between cooperation costs and those in simulation year 200 are shown in Figs 8 and 9. Their average remained almost constant, regardless of the cooperation cost. When the proportion of cooperators was close to 0.5, there was a large standard deviation among the simulation trials, but their averages did not fluctuate much.

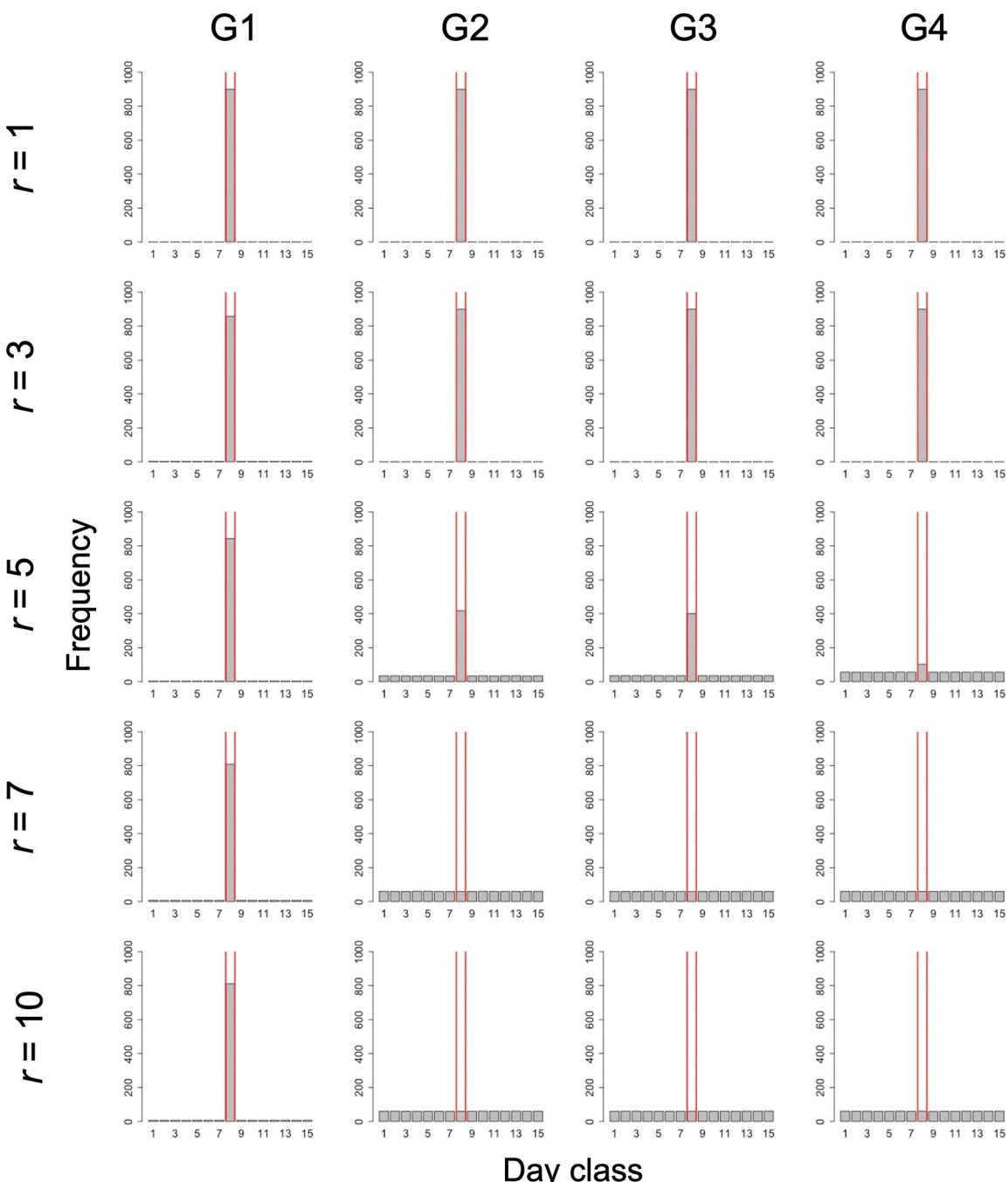

**Fig 5. Frequency distribution of irrigation start dates under the condition of complete information without exogenous preference (scenario 1).** The frequency distribution is indicated by gray bars. The area between the red vertical lines represents the optimal irrigation start date. Columns (G1)–(G4) of the panel show the results for Group Conditions 1–4, respectively. This is the result for year 200 and $C_{co}$ = 10000 in the simulation.

### 3.5. Robustness of simulation results against changes in the average degree

There was a difference in the proportion of cooperators and the standard deviation of irrigation starting dates among the conditions with different average degrees (ranging from 4 to 20) of the farm's information reference network (Figs 10 and 11). This difference was most

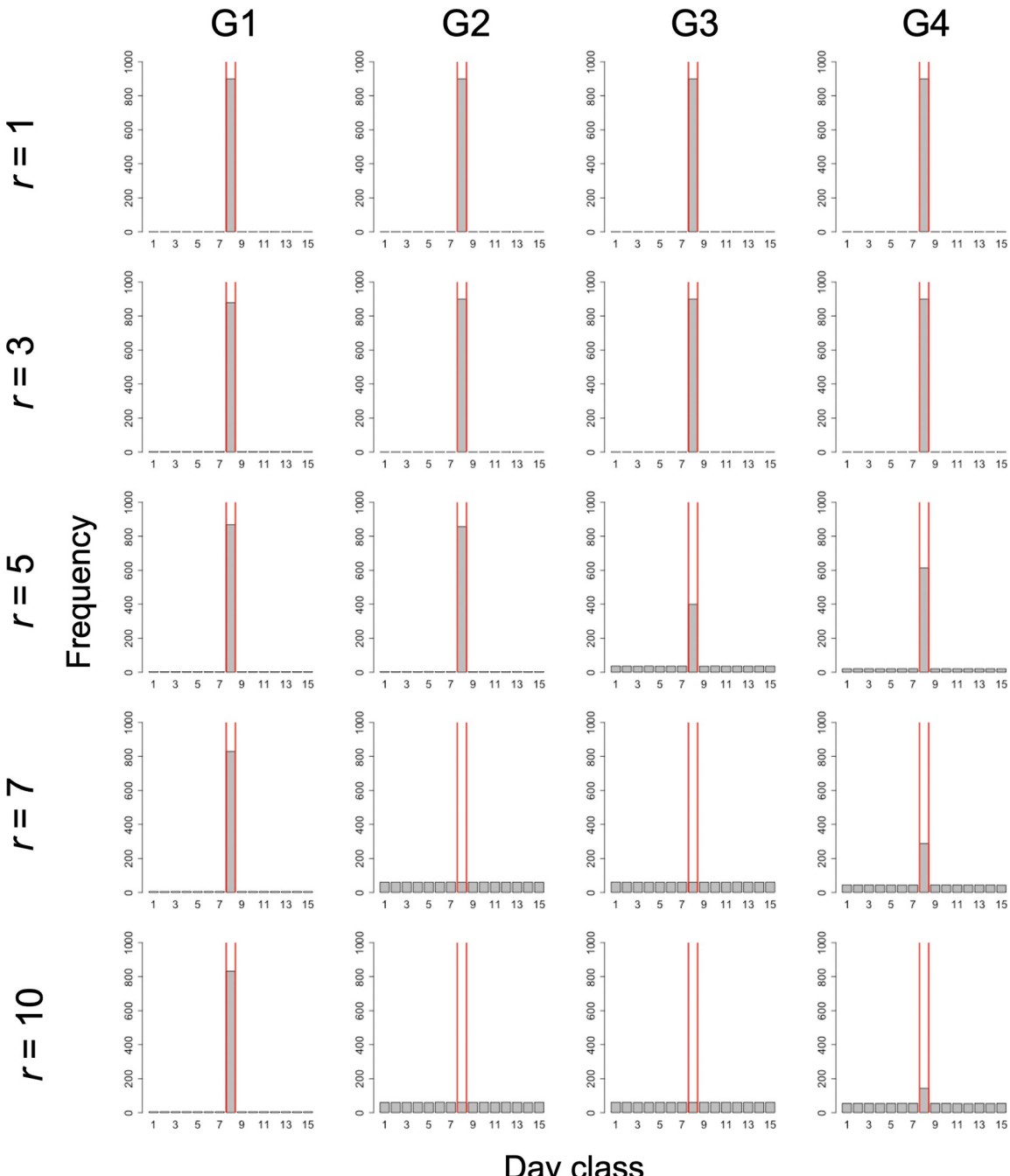

**Fig 6. Frequency distribution of irrigation start dates under the condition of incomplete information without exogenous preference (scenario 2).**

pronounced in condition 4, where the proportion of cooperators and the standard deviation of irrigation starting dates became zero as the average degree increased from 4 to 8 or more.

In contrast, in group conditions 1–3, the effect of the change in the average degree on the proportion of cooperators and the standard deviation of irrigation starting dates was relatively small. In group condition 1, the proportion of cooperators and the standard deviation of irrigation starting dates decreased as the average degree increased. Under the complete information condition of

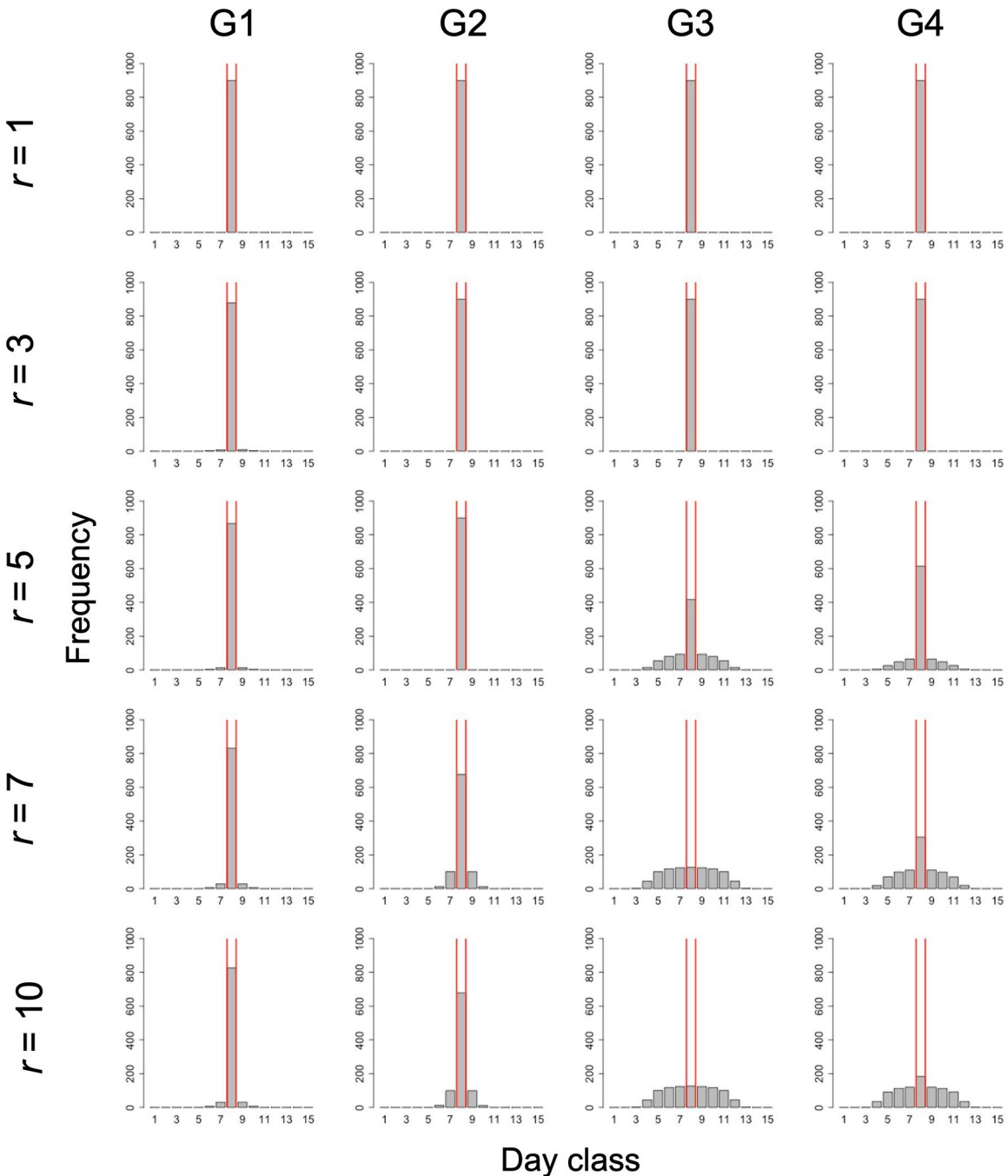

**Fig 7. Frequency distribution of irrigation start dates under the condition of incomplete information with exogenous preference (scenario 3).**

group conditions 2 and 3, the critical multiplication factor decreased as the average degree increased from 4 to 8 or more. Under incomplete information with and without exogenous preference of group conditions 2 and 3, the proportion of cooperators for conditions where $r$ was greater than the critical multiplication factor varied by approximately 20–30% among different average degree conditions. However, the variation among the average degree conditions in the standard deviation of irrigation start dates under this condition of $r$ was very small.

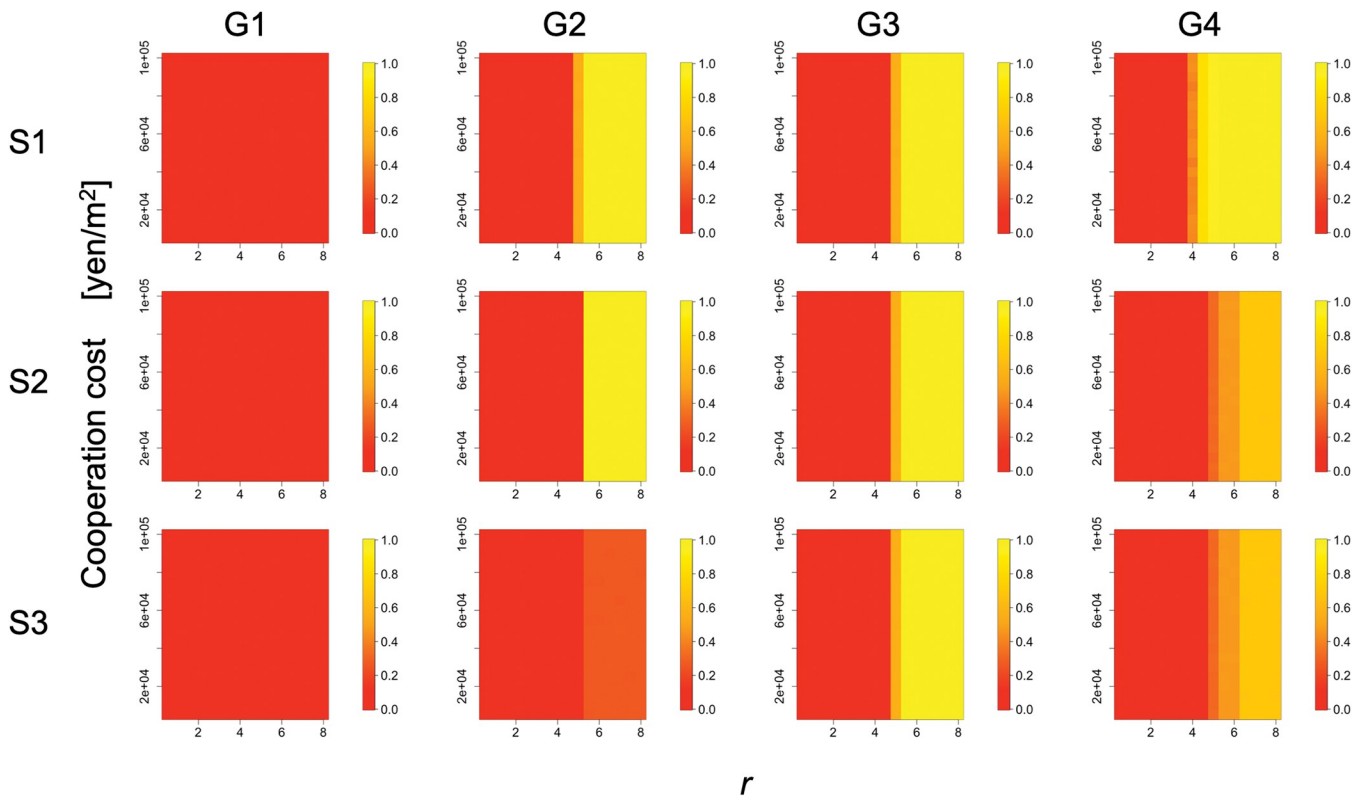

**Fig 8. Relationships among *r*, cooperation costs, and the proportion of cooperators (the average degree is four).** Columns (G1)–(G4) of the panel show the results for group conditions 1–4. Rows (S1)–(S3) of the panel display the condition of complete information without exogenous preference (scenario 1), the condition of incomplete information without exogenous preference (scenario 2), and the condition of incomplete information with exogenous preference (scenario 3), respectively. Color indicates the proportion of cooperators. This is the result for year 200 and $C_{co}$ = 10000 in the simulation.

We performed simulations under the condition that the cooperation cost $C_{co}$, is 10000–100000 [yen/ha/year] when the average degree of the farmer's information reference network is 8, 12, and 20. As a result, despite changes in the cooperation cost, the results for the proportion of cooperators and the standard deviation of irrigation starting dates remained unchanged (see the S1 Appendix for details).

## 4. Discussion

### 4.1. Which of the divided groups and overlapped groups promotes the dispersion of the irrigation start dates?

In this study, we investigated an effective scheme for promoting the dispersion of the irrigation start dates, based on the existing body of knowledge of evolutionary game theory and spatial public goods game. In this section, we discuss the difference in the results of promoting the dispersion of the irrigation start dates brought about by the overlapped and divided (or non-overlapped) groups, which are the essential rules of the spatial public goods game. In real-life settings, agricultural water is often managed and adjusted by divided groups; for example, in Japan, hamlet and water-utilization associations such as land improvement districts and irrigation associations. However, the simulation results of this study showed that the overlapped group conditions promoted the dispersion of the irrigation start dates more than the divided group conditions. From this, we infer that adopting overlapped groups rather than divided groups is more effective in promoting cooperation among farmers and dispersion of the

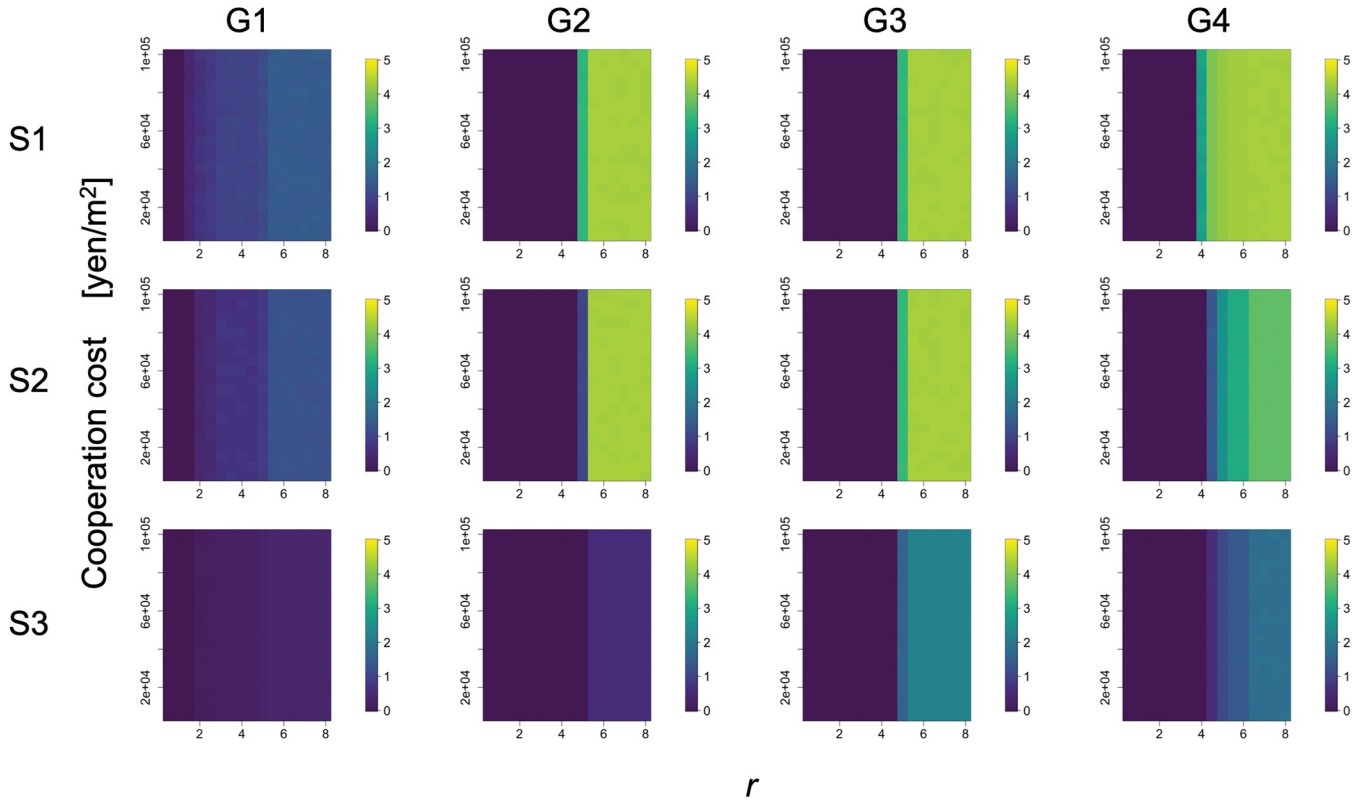

**Fig 9. Relationships among *r*, cooperation costs, and standard deviation of irrigation starting dates (the average degree is four).** Columns (G1)–(G4) of the panel show the results for group conditions 1–4. Rows (S1)–(S3) of the panel display the condition of complete information without exogenous preference (scenario 1), the condition of incomplete information without exogenous preference (scenario 2), and the condition of incomplete information with exogenous preference (scenario 3), respectively. The color indicates the standard deviation of the irrigation start dates. This is the result for year 200 and $C_{co}$ = 10000 in the simulation.

irrigation start dates. In a study of evolutionary games related to farm water intake, Nhim et al. [40] proposed a way whereby cooperators "punish" defectors to promote cooperation. Naka-maru and Yokoyama [41] showed that the rule that a candidate is excluded from membership if group members regard the candidate's reputation as bad (i.e., ostracism) sustains coopera-tion in the public goods game. However, such punishment schemes can induce excessive social sanctions, such as ostracism in rural societies [42], and thus should be implemented carefully. Meanwhile, the scheme we propose avoids the problem of excessive social sanctions, as opposed to punishment schemes.

### 4.2. Comparison of the effects of complete and incomplete (uncertain) information conditions on the dispersion of the irrigation start dates

Even if game theory predicts a diffusion of cooperation, cooperation in real-world settings may not diffuse thoroughly if the following conditions are not met: the central government has accurate information about player behavior by monitoring players and enforcing sanctions on defectors or providing subsidies to cooperators [28]. In addition, obtaining information about a player's behavior is often costly for the government. Ostrom [28] avoided the monitor-ing and control over players' behavior by the central government and proposed a player's self-financed contract-enforcement game as a solution to this problem. In contrast, we proposed the scheme to promote cooperation in public goods games by estimating information about

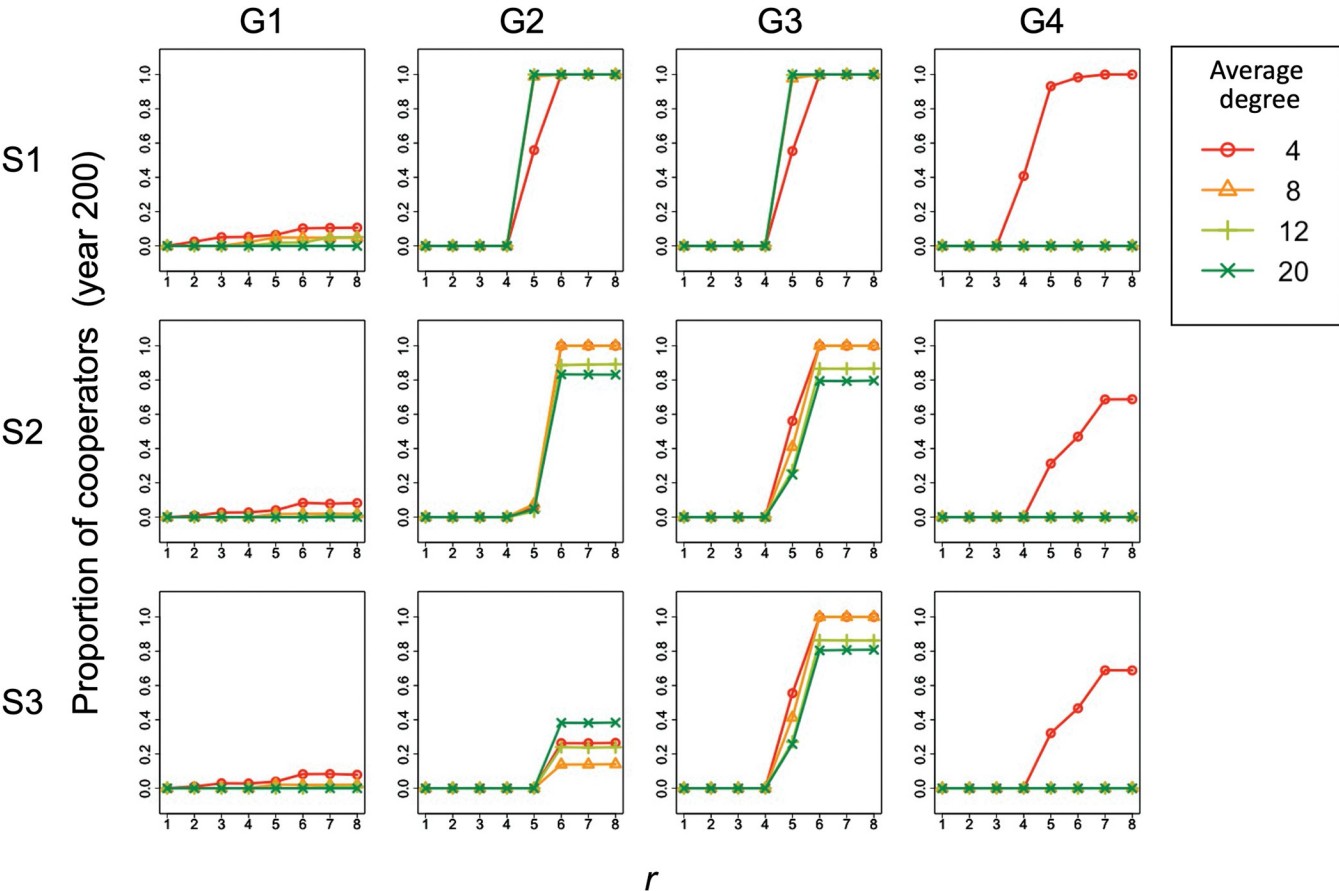

**Fig 10. Relationship between _r_ and the proportion of cooperators under different average degree conditions.** Columns (G1)–(G4) of the panel show the results for group conditions 1–4. Rows (S1)–(S3) of the panel display the condition of complete information without exogenous preference (scenario 1), the condition of incomplete information without exogenous preference (scenario 2), and the condition of incomplete information with exogenous preference (scenario 3), respectively. The shape and color of the plots indicate the average degree of the conditions. This is the result for year 200 and $C_{co}$ = 10000 in the simulation.

players' strategy using the data that governments can collect, such as data on the dispersion of irrigation start date.

In this study's scheme, the government needs to determine the subsidy amount based on the number of cooperating farmers; however, the government does not always have this information. We examined the diffusion of cooperation and the dispersion of irrigation starting dates under both the complete information condition (i.e., the government has complete information about the number of cooperating farmers) and the incomplete information condition (i.e., the government does not have complete information about the number of cooperating farmers). In particular, for the incomplete information condition, we assumed uncertain information on the number of cooperating farmers, estimated from the unevenness of the irrigation start dates within the group. As a result, with complete information, the diffusion of cooperators and the variance of the irrigation start date were the highest in Group Condition 4 among all group conditions. In contrast, in the incomplete (uncertain) information condition, these were the highest in Group Condition 3. This result shows that the scheme to be adopted differs, depending on the uncertainty of the information regarding the number of cooperating farmers.

In recent years, the development of satellite technology has made it possible to monitor irrigation start dates at the level of individual paddy fields, enabling governments to obtain

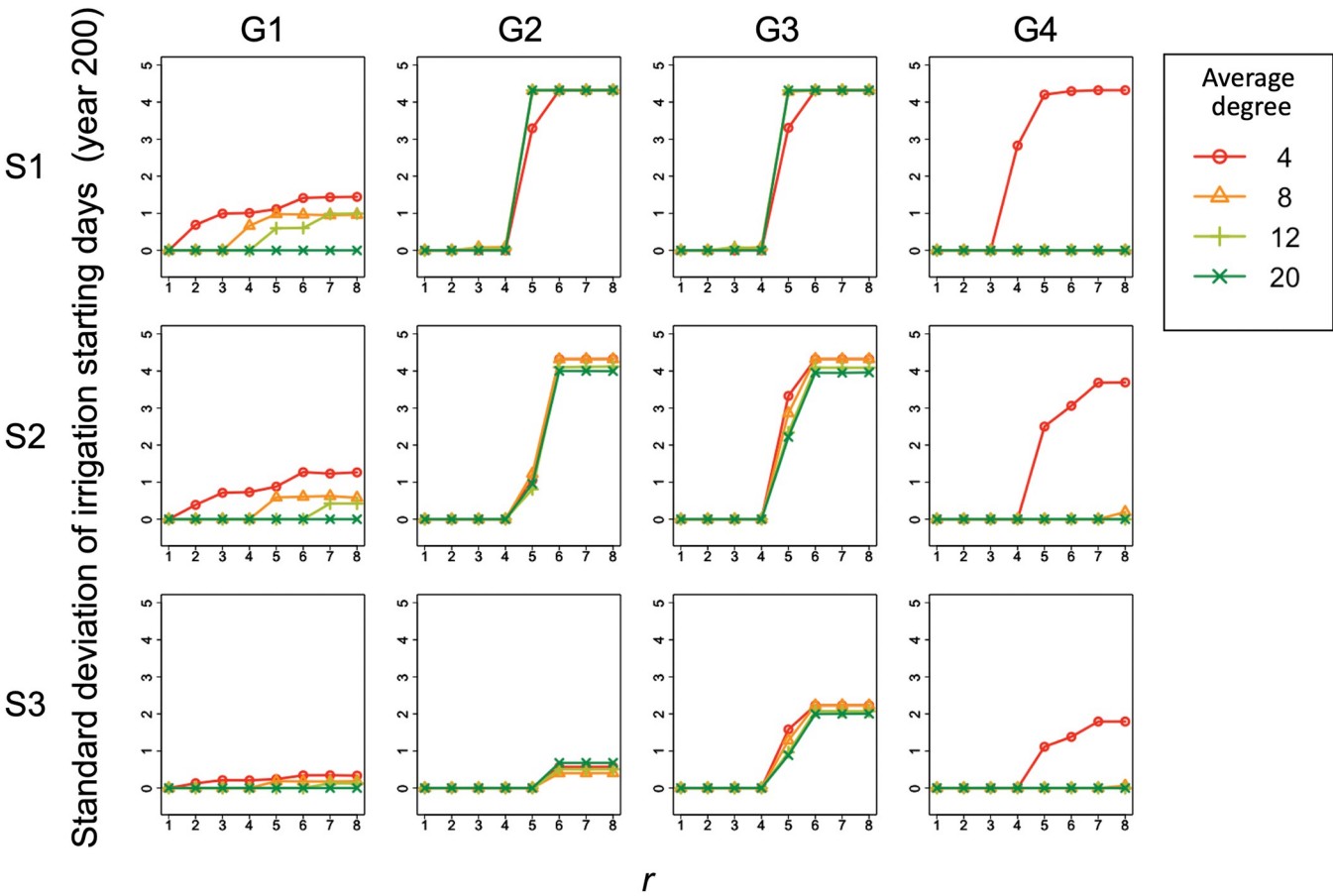

**Fig 11. Relationship between *r* and the standard deviation of irrigation starting dates under different average degree conditions.** Columns (G1)–(G4) of the panel show the results for group conditions 1–4. Rows (S1)–(S3) of the panel display the condition of complete information without exogenous preference (scenario 1), the condition of incomplete information without exogenous preference (scenario 2), and the condition of incomplete information with exogenous preference (scenario 3), respectively. The shape and color of the plots indicate the average degree of the conditions. This is the result for year 200 and $C_{co} = 10000$ in the simulation.

information about the dispersion of irrigation start date based on satellite data [14, 29–32]. Combined use of our proposed scheme and field-level irrigation start date data, obtained by satellites, can increase the number of cooperating farmers and disperse irrigation start dates. In addition, combined use of them can reduce the administrative cost for the government to disperse irrigation start dates. Reducing the costs of adaptation measures is becoming increasingly important. In recent years, when quantifying risks has become possible, it has been recommended that adaptation measures be adopted to examine their cost-effectiveness [43–45]. Moreover, even under serious uncertainty in the risk, the adaptation measure should be implemented only if the outcome of not doing so would be catastrophic, and the measure should be relatively low cost (i.e., the Maximin strategy; [43, 44, 46, 47]). In other words, in both certain and uncertain risk cases, even effective adaptations may not be adopted if the costs of their implementation are very high. Therefore, lowering costs allows for a wider range of measures to be adopted and implemented, thus providing better adaptation to climate change. Furthermore, governments can also obtain information of the number of cooperating farmers by relying on farmers' self-report their cooperation and defection strategies, but farmers may make false reports for their own benefit. The combined use of our proposed scheme satellite data can avoid the problem of false reporting by farmers.

### 4.3. Effect of exogenous preference on the dispersion of irrigation start dates under incomplete (uncertain) information condition

In this study, we investigated the effect of having a preferred irrigation start date—based on factors other than the payoff (e.g., cultural and social factors such as long-standing traditions and religious beliefs)—on the proportion of cooperators and the dispersion of the irrigation start dates.

In the absence of an exogenous preference for the irrigation start date, Group Condition 3 resulted in a higher proportion of cooperators and demonstrated a highly dispersed distribution of irrigation start dates with lower subsidy amounts than Group Condition 4. Note that Group Condition 3 requires farmers to adjust their irrigation start dates with neighboring groups, which does not affect the subsidy they receive. If the cooperating farmer stops engaging in adjusting their irrigation start dates with neighboring groups under Group Condition 3, it will correspond to Group Condition 2. Even when Group Condition 3 shifts to Group Condition 2 due to the absence of this adjustment, a higher proportion of cooperators and a higher dispersion of irrigation start dates can be maintained than in Group Condition 4. Therefore, our results indicate that the scheme of Group Condition 3 should be adopted if farmers in the scheme implementation area do not have an exogenous preference for the irrigation start date.

Moreover, when farmers had exogenous preferences for the irrigation start date, Group Condition 3 resulted in a higher proportion of cooperators and a highly dispersed distribution of irrigation start dates with lower subsidy amounts than Group Condition 4. In this case, too, if the cooperator stops engaging in adjusting their irrigation start dates with neighboring groups under Group Condition 3, it shifts to Group Condition 2. When Group Condition 3 changes to Group Condition 2 due to the absence of this adjustment, the proportion of cooperators and the dispersion of the punishment period are smaller than in Group Condition 4. Therefore, if farmers perform this adjustment, the scheme of Group Condition 3 maximizes the proportion of cooperators and the dispersion of the irrigation start dates; however, if farmers do not perform this adjustment, the scheme under Group Condition 4 maximizes them. This result differs from the case without the exogenous preference. In conclusion, if farmers in the scheme implementation area have an exogenous preference, the government needs to choose an appropriate scheme type based on the sameness/difference between the groups assumed by farmers and governments.

### 4.4. Robustness of the results of this study

In this study, the results of the number of cooperators and the dispersion of the irrigation start dates did not changed as cooperation costs ($C_{co}$) variated (range: 1–6 times). This means that the results presented in this study would hold, even if there are relatively large fluctuations in the cooperation cost. At the same time, this also means that the results presented in this study would hold, even if there are relatively large fluctuations in the parameter $\kappa$ associated with updating farmers' strategies. it is because, according to equations (1)–(8), in agent-based model we proposed, the simulation outcomes when the cooperation cost is multiplied by $k$ are completely the same as the outcomes when $\kappa$ is multiplied by $1/k$.

However, changes in the average degree of farmers' information reference networks can affect the proportion of cooperating farmers and the dispersion of irrigation start dates. In particular, in group condition 4, the increase in the average degree made all farmers defectors, and the dispersion in irrigation start dates disappeared. This result indicates that the government should avoid adopting group condition 4 when the average degree is large. In contrast, for group conditions 1–3, the influence of the change in the average degree on the proportion of cooperating farmers and the dispersion of the irrigation start dates was relatively small. In

particular, when subsidy amounts are calculated in one subsidy group, such as under group conditions 2 and 3, the average degree conditions have little effect on cooperation promotion, which is consistent with the results of Szolnoki and Perc [18]. In other words, they found that differences in the set of potential strategy donors did not affect the critical multiplication factor in the spatial public goods game when the group size sharing the profits among players is small i.e., less than about 20 players in the group (note that the strategy donor in their study corresponds to the reference farmer in this study). As a side note, they reported that the critical multiplication factor is relatively large as the average degree increases when the group size sharing profits among players is large. Applying this result to the present study, when the size of the subsidy group is large, it is expected that cooperation will be less likely to be promoted, and the dispersion of the irrigation start dates will decrease as the average degree increases. Finally, comparing the results among different group conditions when the average degree is greater than 4, group condition 3 can achieve the highest proportion of cooperating farmers and the highest dispersion of the irrigation start dates under conditions of a high average degree.

## 4.5. Mechanism by which differences in group conditions influence the promotion of cooperation

To understand intuitively why cooperation or defection is more dominant in a population depending on the group condition, we interpret this reason based on the relationship between the payoffs of cooperative and defective farmers who are directly connected by links in the square lattice. Hereafter, the neighboring cooperating and defecting farmers are referred to as C and D, respectively.

In a spatial public goods game, there is no group within the population (it can be said that the entire population is one group), and "payoff of C < payoff of D" always holds [19, 48]. Therefore, all players defect in the population. However, when there are multiple groups in a population, regardless of whether the groups are divided (non-overlap) or overlap, this inequality does not always hold.

First, we consider the inequality between these payoffs under the divided group condition. In this case, adjacent players can be distinguished as belonging to the same group or a different group. "Payoff of C < payoff of D" always holds when C and D belong to the same group, but "payoff of C > payoff of D" can also hold when C and D belong to different groups (Note that the network also has links between players belonging to different groups, so players can also reference the payoff of a player belonging to a different group). For example, when there is a group where all players are D and a group where all players are C, the players in the latter will receive more subsidies than those in the former, so "payoff of C > payoff of D" holds between adjacent players in different groups. Specifically, when there is a group in which all players are D and a group in which all players are C, from Eqs (1) and (2), the payoff of C is $R_{opt}-C_{opt}+C_{co}r-C_{co}$, and the payoff of D is $R_{opt}-C_{opt}$. Therefore, the condition for "payoff of C > payoff of D" is $r > 1$. Furthermore, when one group consists of $n$ players in C and $5-n$ players in D and another group consists of $m$ players in C and $5-m$ players in D, by using the same method described above, the inequality of the payoffs of C and D depending on the condition of $r$ can be derived.

Now, we consider overlapped group condition. Under this condition, unlike in the previous divided group conditions, adjacent players always belong to the same group. In this condition, there is a possibility that "payoff of C > payoff of D" holds between all adjacent players. Specifically, we consider the payoffs of adjacent C and D when the groups overlap. For simplicity, the payoffs that comes from the cooperation of players other than target players (i.e., C and D) are assumed to be equal for C and D. From Eqs (6) and (7), the payoff of C is $R_{opt}-C_{opt}+C_{co}r/5$

$-C_{co}+M$, and the payoff of D is $R_{opt}-C_{opt}+2C_{co}r/25+M$. $M$ is the payoff that comes from the cooperation of players other than the target players. Therefore, when groups overlap, the condition for "payoff of C > payoff of D" to hold is $r>25/3$. In other words, "payoff of C < payoff of D" holds between all adjacent players when the subsidy amount is small, but "payoff of C > payoff of D" holds between them when subsidy amount is large.

In this way, when the group is divided, even if the subsidy amount increases, "payoff of C > payoff of D" does not hold between adjacent players in the same group, and "payoff of C > payoff of D" holds only between adjacent players in different groups. However, if the groups overlap, increasing the subsidy amount results in "payoff of C > payoff of D" between adjacent players. This is the reason overlapped group conditions results in a higher proportion of cooperators with fewer subsidy amounts than divided group conditions.

## 4.6. Limitations of this study

The agent-based model developed in this study follows Axelrod's KISS principle, and some elements of the target phenomenon were omitted for simplicity. We present a supplementary consideration of the effects of the omitted elements below.

First, in real-life settings, when water resources are not abundant and if farmers' water use is concentrated in terms of time, their available water may decrease, thereby decreasing their yields and sales. Here, we can regard water resources as competitive resources. However, for simplicity, we assumed that water is a non-competitive resource in this study. When water is assumed to be a competitive resource, we can achieve a larger dispersion of the irrigation start dates by using lower subsidy amounts compared to the findings obtained in this study. This is because farmers' payoffs decrease due to water shortages as the irrigation start dates are concentrated, and they have a greater incentive to disperse the irrigation start dates.

Although this study assumes that sales and costs per unit of cultivated areas are constant, efficiency may increase and costs may decrease as the cultivated area of an individual farmer increases. In such cases, the results may change, depending on the frequency distribution of the cultivated areas of individual farmers and the spatial arrangement of farmers with different cultivated areas on the square lattice. However, further research is required to confirm this.

In this study, we assumed that the relationships among farmers are regular graphs (i.e., farmers have the same degree). However, farmer relationships in real-life settings are complex. Previous studies of spatial public goods games have shown the diffusion of cooperators can be further promoted by changing the relationship assumption from regular graphs to more complex network structures such as scale-free networks and networks with high clustering coefficients [17, 49, 50]. Therefore, in our proposed scheme, it may be possible to promote the diffusion of cooperation and dispersion of irrigation start dates by applying a more complex network.

## Supporting information

**S1 Appendix.**
(DOCX)

**S1 Data.**
(ZIP)

## Author Contributions

**Conceptualization:** Yoshiaki Nakagawa.

**Formal analysis:** Yoshiaki Nakagawa.

**Funding acquisition:** Masayuki Yokozawa.

**Investigation:** Yoshiaki Nakagawa.

**Methodology:** Yoshiaki Nakagawa.

**Project administration:** Masayuki Yokozawa.

**Visualization:** Yoshiaki Nakagawa.

**Writing – original draft:** Yoshiaki Nakagawa.

**Writing – review & editing:** Masayuki Yokozawa.

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
