## [Decision Letter · Decision Letter 0]

19 Dec 2022

PONE-D-22-32008A social system to disperse the irrigation start date based on the spatial public goods gamePLOS ONE

Dear Dr. Nakagawa,

Thank you for submitting your manuscript to PLOS ONE. After careful consideration, we feel that it has merit but does not fully meet PLOS ONE’s publication criteria as it currently stands. Therefore, we invite you to submit a revised version of the manuscript that addresses the points raised during the review process.

We look forward to receiving your revised manuscript.

Kind regards,

Jun Tanimoto

Academic Editor

PLOS ONE

“This research was supported by the Environment Research and Technology Development Fund (JPMEERF20S11805) of the Environmental Restoration and Conservation Agency of Japan.”

“Initials of the authors who received each award:

MY

Grant numbers awarded to each author:

JPMEERF20S11805

The full name of each funder:

the Environment Research and Technology Development Fund (the Environmental Restoration and Conservation Agency of Japan)

URL of each funder website:

https://www.erca.go.jp/erca/english/index.html

Did the sponsors or funders play any role in the study design, data collection and analysis, decision to publish, or preparation of the manuscript?:

Reviewers' comments:

Reviewer's Responses to Questions

**Comments to the Author**

1. Is the manuscript technically sound, and do the data support the conclusions?

Reviewer #1: Partly

Reviewer #2: Partly

2. Has the statistical analysis been performed appropriately and rigorously? 

Reviewer #1: N/A

Reviewer #2: N/A

3. Have the authors made all data underlying the findings in their manuscript fully available?

Reviewer #1: Yes

Reviewer #2: Yes

4. Is the manuscript presented in an intelligible fashion and written in standard English?

Reviewer #1: No

Reviewer #2: Yes

5. Review Comments to the Author

Reviewer #1: This work establishes a quite specific spatial version PGG, and report its numerical results obtained thru a series of MASs. Honestly, there is none of scientifically new and novel point, simply because such a fundamental Spatial PGGs have been well-cultivated in the field of social physics, nonlinear science, applied mathematics, theoretical biology, and other neighboring areas including information science as well as applied mathematics, which the authors have hardly reviewed the holistic picture of those precursors’ works; that are quite rich, and what’s going on the front line in such fields whatsoever. Hence, it’s difficult to positively evaluate this contribution from stringent scientific standpoints. Yet, I still have somewhat positive impression from this contribution, since the authors have been inspired by a strong social motivation, which comes from the background aiming an application of agriculture topics; especially concerned on a typical social dilemma situation, where an egocentric farmer not cooperating to time-sharing irrigation scheme competes with an altruistic cooperator. Which might be somehow meaningful to the audience as an example (but rather specific) of application based on PGG.

Although I’m with a certain positiveness, I do believe that the authors must improve this contribution to more impressive and attractive to the audience.

As below, I give several suggestions which should be sincerely responded in the rebuttal materials and should be reflected in the revised MS.

#1.

Section 2.2 defines the payoff structure of their specific PGG. And, as Section 2.3 claiming, they relied on k=4 lattice as underlying topology and PW-Fermi as strategy updating rule. Although, at a glance, the game structure seems a bit complicated, it could be simplified so as to put their model onto a counterpart theoretical model. Why didn’t they visit such a theoretical approach, which is more transparent to quantitively discuss about the dynamical feature of their model. The authors should justify this by citing some literatures giving the theoretical framework for multi-players game; perhaps recent books such as; Sociophysics Approach to Epidemics, Springer, 2021.

#2.

They presumed the number of neighbors is fixed at 4. Generally speaking, and as an appropriate theorical prediction easily suggesting, the network reciprocity becomes difficult to be emerged with increase of degree. Why didn’t they explored other cases than k=4 to deepen their discussion?

#3.

Visual results look meager. To as to attract wider audience, they should illustrate more impressive result. For instance, instead of bunch of line graphs they displayed, heat-maps of cooperation fraction along Cooperation cost and r should be provided.

#4.

Following to the previous Item; #3, a general PGG can be accounted by a universal dilemma strength, which depends on both the cooperation cost and the amplification factor; r in their model, which can be called as ‘dilemma weakness’. They should deepen this point in Discussion part. For their guidance, they should visit the concept of universal dilemma strength for 2 by 2 games and the concept of Social Efficiency Deficit, recently reported. They should refer those by citing relevant literatures; (i) Universal scaling for the dilemma strength in evolutionary games, Physics of Life Reviews 14, 1-30, 2015, (ii) Scaling the phase- planes of social dilemma strengths shows game-class changes in the five rules governing the evolution of cooperation, Royal Society Open Science, 181085, 2018, (iii) Social efficiency deficit deciphers social dilemmas, Scientific Reports 10, 16092, 2020.

Reviewer #2: The authors studied the application of PGG game on irrigation start date by considering several factors, applying agent-based simulations. The authors gave some simulated result where square lattice presumed for underlying network, which ensures that the newly proposed strategy updating rule based on dynamic switching PW-Fermi. Such presented result seems somehow informative. In sum, I can evaluate this work can be published on PLOSE. Yet, to improve the contents ensuring more impressive information to the audience of PLOSE one, I would like to give following point to be revised in the final MS.

##1. The current form of introduction part is too large, some text is unclear to understand with some incorrect descriptions and there have some information without citing any previous works. Please rewrite the introduction to make it concise and informative.

##2. I could not find any substantial discussion that can fully reflect the assumption of PGG model setup (for example pairwise game). Is it possible to express current model by using pairwise (two by two) evolutionary game mode? Please also mention clearly about all formulation and parameter settings including assumed values that can be fully understandable.

##3. The results seem less impressive and insufficient. I think, it will be very meaningful if authors plot some 2D phase diagram varying two parameters (r versus cost) or others. By introducing 2D heatmap can be explore details explanation of current works.

##4. Authors can introduce social efficiency deficit, and dilemma strength to make this work more realistic and scientifically interesting by following some previous researches:

Influence of bolstering network reciprocity in the evolutionary spatial Prisoner’s Dilemma game: A perspective, European Physical Journal B 91, 312, 2018.

Dilemma strength as a framework for advancing evolutionary game theory: “Universal scaling for the dilemma strength in evolutionary games”, Physics of Life Reviews 14, 56-58, 2015.

“Do humans play according to the game theory when facing the social dilemma situation?” A survey study, EVERGREEN, 07(01), 7-14 (2020).

Social efficiency deficit deciphers social dilemmas, scientific reports (Nature), 10, 16092 (2020).

Tanimoto; Evolutionary Games with Sociophysics: Analysis of Traffic Flow and Epidemics, Springer, 2019

6. PLOS authors have the option to publish the peer review history of their article (what does this mean?). If published, this will include your full peer review and any attached files.

Reviewer #1: No

Reviewer #2: No

---

## [Author Response · Author response to Decision Letter 0]

5 May 2023

Italics are editor & referee comments followed by our responses in regular type.

< Changes in Figures and Appendix >

In the revised manuscript, we made the following changes to figures and appendix. 

We have added new Figures 2, 3, 10, and 11 in the revised manuscript.

The previous Figures 1–5 are now Figures 1, 4–7 in the revised manuscript, respectively.

The previous Figures 6 and 7 have been replaced by Figures 8 and 9 in the revised manuscript, respectively.

We have newly added Appendix.

Comments from reviewers:

Reviewer#1: 

This work establishes a quite specific spatial version PGG, and report its numerical results obtained thru a series of MASs. Honestly, there is none of scientifically new and novel point, simply because such a fundamental Spatial PGGs have been well-cultivated in the field of social physics, nonlinear science, applied mathematics, theoretical biology, and other neighboring areas including information science as well as applied mathematics, which the authors have hardly reviewed the holistic picture of those precursors’ works; that are quite rich, and what’s going on the front line in such fields whatsoever. Hence, it’s difficult to positively evaluate this contribution from stringent scientific standpoints. Yet, I still have somewhat positive impression from this contribution, since the authors have been inspired by a strong social motivation, which comes from the background aiming an application of agriculture topics; especially concerned on a typical social dilemma situation, where an egocentric farmer not cooperating to time-sharing irrigation scheme competes with an altruistic cooperator. Which might be somehow meaningful to the audience as an example (but rather specific) of application based on PGG.

Although I’m with a certain positiveness, I do believe that the authors must improve this contribution to more impressive and attractive to the audience.

As below, I give several suggestions which should be sincerely responded in the rebuttal materials and should be reflected in the revised MS.

Comment 1

Section 2.2 defines the payoff structure of their specific PGG. And, as Section 2.3 claiming, they relied on k=4 lattice as underlying topology and PW-Fermi as strategy updating rule. Although, at a glance, the game structure seems a bit complicated, it could be simplified so as to put their model onto a counterpart theoretical model. Why didn’t they visit such a theoretical approach, which is more transparent to quantitively discuss about the dynamical feature of their model. The authors should justify this by citing some literatures giving the theoretical framework for multi-players game; perhaps recent books such as; Sociophysics Approach to Epidemics, Springer, 2021.

Response to comment 1 (Reviewer #1): 

>Simplification to put model onto a counterpart theoretical model

The results of a simpler theory corresponding to the model of this study have already been indicated by previous studies (Szabo and Hauert, 2002; Haurt and Szabo, 2003; Brandt et al., 2003; Santos et al., 2008; Yang et al., 2009; Helbing et al., 2010; Perc et al., 2013). These studies are cited in the revised manuscript (Section 1 in the revised manuscript). However, this is not the focus of our research. Our research interest is to apply those findings to an actual social problem (water shortage during the puddling season). When applying the theoretical findings of previous research to actual social problems, it is important to consider the situation in which the government that pays the payoff (or subsidy) cannot directly know the parameters in game theory. In practice, the lack of information on these parameters causes a divergence between theoretical and real-world results. That is, cooperation cannot be promoted in reality while it is possible to promote cooperators in theory (Ostrom,1990). To consider a more realistic situation, our study focused on the situation in which subsidizing governments, which subsidize farmers, do not have direct knowledge of the number of cooperators (or defectors) in each group (i.e., the scenarios 2 and 3 in the revised manuscript). In this study, the government statistically estimates the number of cooperators (or defectors) in each group from the distribution of irrigation start dates and determines subsidies depending on the estimated number of cooperators. In addition, when considering applications to practical social problems, it is necessary to consider possible disturbing factors. For this reason, in a more realistic scenario (Scenario 3), we also examine the results when sociocultural external influence in the selection of irrigation start dates. For the purposes described above, practical application-oriented simulations are considered more suitable than simple theoretical models. To provide readers with a clearer understanding of this, we have rewritten the introduction (Section 1 in the revised manuscript) and added the schematic of the simulation (Section 2.1 and Figure 2 in the revised manuscript).

>More transparent Theoretical approach to quantitively discuss the dynamical feature of their model

We are still unsure exactly what entails a more transparent theoretical approach. Please be specific about what this approach is. After reading the recommended book (Sociophysics Approach to Epidemics, Springer, 2021), we thought it was about the social efficient deficit (SED), the multiplayer game version of dilemma strength (DS). The response about the SED is answered in response to comment 4 (Reviewer #1).

Comment 2.

They presumed the number of neighbors is fixed at 4. Generally speaking, and as an appropriate theorical prediction easily suggesting, the network reciprocity becomes difficult to be emerged with increase of degree. Why didn’t they explored other cases than k=4 to deepen their discussion?

Response to comment 2 (Reviewer #1): 

We calculated the results for different degree conditions and modified the manuscript. Previous studies reported that the average degree of the farmer's network in the farmer's information sharing network ranged from 4–19 (Isaac et al., 2007; Saint Ville et al., 2016; Beaman and Dillon, 2018). Therefore, we newly calculated cases where the average degree is 8, 12, and 20 and added the method, result, and discussion under the condition of the different average degrees to the revised manuscript (Sections 2.7, 3.5, and 4.4 in the revised manuscript).

Comment 3

Visual results look meager. To as to attract wider audience, they should illustrate more impressive result. For instance, instead of bunch of line graphs they displayed, heat-maps of cooperation fraction along Cooperation cost and r should be provided.

Response to comment 3 (Reviewer #1): 

Figures 6 and 7 have been recreated as heatmaps in the revised manuscript.

Comment 4

Following to the previous Item; #3, a general PGG can be accounted by a universal dilemma strength, which depends on both the cooperation cost and the amplification factor; r in their model, which can be called as ‘dilemma weakness’. They should deepen this point in Discussion part. For their guidance, they should visit the concept of universal dilemma strength for 2 by 2 games and the concept of Social Efficiency Deficit, recently reported. They should refer those by citing relevant literatures; (i) Universal scaling for the dilemma strength in evolutionary games, Physics of Life Reviews 14, 1-30, 2015, (ii) Scaling the phase- planes of social dilemma strengths shows game-class changes in the five rules governing the evolution of cooperation, Royal Society Open Science, 181085, 2018, (iii) Social efficiency deficit deciphers social dilemmas, Scientific Reports 10, 16092, 2020.

Universal scaling for the dilemma strength in evolutionary games, Physics of Life Reviews 14, 1-30, 2015, 

Scaling the phase- planes of social dilemma strengths shows game-class changes in the five rules governing the evolution of cooperation, Royal Society Open Science, 181085, 2018, 

Social efficiency deficit deciphers social dilemmas, Scientific Reports 10, 16092, 2020.

Response to comment 4 (Reviewer #1): 

According to the book (Sociophysics Approach to Epidemics, Springer, 2021) and these papers, dilemma strength can only be defined for 2×2 games and cannot be defined for multiplayer games, including public goods games (PGG). According to this literature, the social efficiency deficit (SED) can be applied to PGG as a measure of a social dilemma. SED is described as the difference between an actor’s average payoff in a social optimum and an evolutionary equilibrium (Arefin et al., 2020; Kabir et al., 2021). Social optimum means the situation in which the actor’s average payoff is maximized. Therefore, SED indicates an improvement in an actor's average payoff from evolutionary equilibrium to the social optimum. When SED is zero, there is no social dilemma; when SED is positive, there is a social dilemma. This metric is remarkably useful for policymakers to solve social dilemmas or aim for social conditions that maximize the sum (or average) of actors' payoffs.

However, our study does not aim to maximize the average of payoffs of actors (farmers). It aims to maximize the standard deviation of the irrigation start dates with a smaller subsidy amount. To maximize the standard deviation of irrigation start dates, not everyone needs to be a cooperator and thus the average of the payoffs of the actors does not need to be maximized. Therefore, for this purpose, it is useful to examine the relationships between the parameter r (or cost) and the standard deviation of the irrigation start date (Figure 9), rather than the relationship between the parameter r (or cost) and SED. We had no idea how using SED could contribute to our goal (i.e., maximizing the standard deviation of irrigation start dates). However, if our understanding of SED is incorrect and is a useful metric for our study, please point it out.

 

Reviewer #2: 

The authors studied the application of PGG game on irrigation start date by considering several factors, applying agent-based simulations. The authors gave some simulated result where square lattice presumed for underlying network, which ensures that the newly proposed strategy updating rule based on dynamic switching PW-Fermi. Such presented result seems somehow informative. In sum, I can evaluate this work can be published on PLOSE. Yet, to improve the contents ensuring more impressive information to the audience of PLOSE one, I would like to give following point to be revised in the final MS.

Comment 1

The current form of introduction part is too large, some text is unclear to understand with some incorrect descriptions and there have some information without citing any previous works. Please rewrite the introduction to make it concise and informative.

Response to comment 1 (Reviewer #2): 

We have revised the introduction (Section 1 in the revised manuscript). We have removed some redundant and unnecessary parts of the introduction and have also rewritten it. We have also increased the citations we consider necessary. If there are still incorrect descriptions, please point them out specifically and we will correct them.

Comment 2

I could not find any substantial discussion that can fully reflect the assumption of PGG model setup (for example pairwise game). Is it possible to express current model by using pairwise (two by two) evolutionary game mode? Please also mention clearly about all formulation and parameter settings including assumed values that can be fully understandable.

Response to comment 2 (Reviewer #2): 

Spatial public goods games are group interactions rather than pairwise interactions like the prisoner's dilemma game. Therefore, it cannot be expressed in the form of a 2-player, 2-strategy (2×2). The payoff is defined by Eqs [Disp-formula pone.0286127.e001]–[Disp-formula pone.0286127.e008] (in Section 2.3 in the revised manuscript) for each group condition. Here, payoffs are defined based on the number of cooperators (or defectors) in each group. In addition, to make our proposed model easier for readers to understand, we added a more detailed description of the model outline (Section 2.1 in the revised manuscript).

Comment 3

The results seem less impressive and insufficient. I think, it will be very meaningful if authors plot some 2D phase diagram varying two parameters (r versus cost) or others. By introducing 2D heatmap can be explore details explanation of current works.

Response to comment 3 (Reviewer #2): 

Figures 6 and 7 have been recreated as heat maps in the revised manuscript.

Comment 4

Authors can introduce social efficiency deficit, and dilemma strength to make this work more realistic and scientifically interesting by following some previous researches:

Influence of bolstering network reciprocity in the evolutionary spatial Prisoner’s Dilemma game: A perspective, European Physical Journal B 91, 312, 2018.

Dilemma strength as a framework for advancing evolutionary game theory: “Universal scaling for the dilemma strength in evolutionary games”, Physics of Life Reviews 14, 56-58, 2015.

“Do humans play according to the game theory when facing the social dilemma situation?” A survey study, EVERGREEN, 07(01), 7-14 (2020).

Social efficiency deficit deciphers social dilemmas, scientific reports (Nature), 10, 16092 (2020).

Tanimoto; Evolutionary Games with Sociophysics: Analysis of Traffic Flow and Epidemics, Springer, 2019

Response to comment 4 (Reviewer #2): 

Please refer to the response to comment 4 (Reviewer #1).

References of our comments

Arefin MR, Kabir KMA, Jusup M, Ito H, Tanimoto J. Social efficiency deficit deciphers social dilemmas. Sci. Rep. 2020; 10: 16092.

Beaman L, Dillon A. Diffusion of agricultural information within social networks: Evidence on geneder in equalities from Mali. J. Dev. Econ. 2018; 133: 147–161.

Brandt H, Hauert C, Sigmund K. Punishment and reputation in spatial public goods games. Proc. R. Soc. London, Ser B 2003; 270: 1099.

Hauert C, Szabo G. Prisoner's dilemma and public goods games in different geometries: Compulsory versus voluntary interactions. Complexity 2003; 8(4): 31–38.

Helbing D, Szolnoki, A, Perc M, Szabó G. Punish, but not too hard: how costly punishment spreads in the spatial public goods game. New J. Phys. 2010; 12: 083005.

Isaac ME, Erickson BH, Quashie-Sam SJ, Timmer VR. Transfer of knowledge on Agroforestry management practices: the structure of farmer advice networks. Ecol. Soc. 2007; 12: 32.

Kabir KMA, Risa T, Tanimoto J. 2021. Prosocial behavior of wearing a mask during an epidemic: an evolutionary explanation. Sci. Rep. 2021; 11: 12621.

Ostrom E. Governing the commons: The evolution of institutions for collective action. Cambridge, New York: Cambridge University Press; 1990.

Perc M, Gómez-Gardenes J, Szolnoki A, Floría L M, Moreno Y. Evolutionary dynamics of group interactions on structured populations: a review. J R Soc Interface. 2013; 10(80): 20120997.

Saint Ville AS, Hickey GM, Phillip LE. Exploring the role of social capital in influence knowledge flows and innovation in smallholder darming communities in the Caribbean. Food. Soc. 2016; 8: 535–549. 

Santos FC, Santos MD, Pacheco JM. Social diversity promotes the emergence of cooperation in public goods games. Nature 2008; 454: 213–216.

Szabó G, Hauert C., Phase transitions and volunteering in spatial public goods games. Phys. Rev. Lett. 2002; 89(11): 118101.

Yang H-X, Wang W-X, Wu Z-X, Lai Y-C, Wang B-H. Diversity-optimized cooperation on complex networks. Phys. Rev. E 2009; E 79: 056107.

---

## [Decision Letter · Decision Letter 1]

10 May 2023

A social system to disperse the irrigation start date based on the spatial public goods game

PONE-D-22-32008R1

Dear Dr. Nakagawa,

We’re pleased to inform you that your manuscript has been judged scientifically suitable for publication and will be formally accepted for publication once it meets all outstanding technical requirements.

Kind regards,

Jun Tanimoto

Academic Editor

PLOS ONE

Additional Editor Comments (optional):

Reviewers' comments:

Reviewer's Responses to Questions

**Comments to the Author**

1. If the authors have adequately addressed your comments raised in a previous round of review and you feel that this manuscript is now acceptable for publication, you may indicate that here to bypass the “Comments to the Author” section, enter your conflict of interest statement in the “Confidential to Editor” section, and submit your "Accept" recommendation.

Reviewer #1: All comments have been addressed

Reviewer #2: (No Response)

2. Is the manuscript technically sound, and do the data support the conclusions?

Reviewer #1: Yes

Reviewer #2: Yes

3. Has the statistical analysis been performed appropriately and rigorously? 

Reviewer #1: Yes

Reviewer #2: Yes

4. Have the authors made all data underlying the findings in their manuscript fully available?

Reviewer #1: Yes

Reviewer #2: (No Response)

5. Is the manuscript presented in an intelligible fashion and written in standard English?

Reviewer #1: Yes

Reviewer #2: (No Response)

6. Review Comments to the Author

Reviewer #1: The responses from the authors seems adequately persuasive to solve my points of inquiry. Hence, now I would like to suggest Accept...

Reviewer #2: (No Response)

7. PLOS authors have the option to publish the peer review history of their article (what does this mean?). If published, this will include your full peer review and any attached files.

Reviewer #1: No

Reviewer #2: No

---

## [Editor Report · Acceptance letter]

16 May 2023

PONE-D-22-32008R1 

A social system to disperse the irrigation start date based on the spatial public goods game 

Dear Dr. Nakagawa:

I'm pleased to inform you that your manuscript has been deemed suitable for publication in PLOS ONE. Congratulations! Your manuscript is now with our production department. 

Kind regards, 

on behalf of

Prof. Jun Tanimoto 

Academic Editor

PLOS ONE